



**Development and Comparison of Empirical Models for All-sky Downward**
**Longwave Radiation Estimation at the Ocean Surface Using Long-term**
**Observations**
**Jianghai Peng[1,2,3,4], Bo Jiang[1,2]\*, Hui Liang[1,2], Shaopeng Li[1,2], Jiakun Han[1,2], Thomas C.**
**Ingalls[3,4], Jie Cheng[1,2], Yunjun Yao[1,2], Kun Jia[1,2], and Xiaotong Zhang[1,2]**
[1]The State Key Laboratory of Remote Sensing Science, jointly sponsored by Beijing Normal University and
Institute of Remote Sensing and Digital Earth of Chinese Academy of Sciences, Beijing, China.
[2]China and Beijing Engineering Research Center for Global Land Remote Sensing Products, Institute of
Remote Sensing Science and Engineering, Faculty of Geographical Science, Beijing Normal University,
Beijing, China.
[3]Center for Global Discovery and Conservation Science, Arizona State University, Tempe, AZ, 85281, USA.
[4]School of Earth and Space Exploration, Arizona State University, Tempe, AZ, 85281, USA.
Corresponding author: Bo Jiang(bojiang@bnu.edu.cn)



**Abstract**

The ocean-surface downward longwave radiation ($R_l$) is one of the most fundamental
components of the radiative energy balance, and it has a remarkable influence on air–sea
interactions. Because of various shortcomings and limits, a lot of empirical models were
established for ocean-surface $R_l$ estimation for practical applications. In this paper, based on
comprehensive measurements collected from 65 moored buoys distributed across global seas
from 1988 to 2019, a new model for estimating the all-sky ocean-surface $R_l$ at both hourly and
daily scales was built. The ocean-surface $R_l$ was formulated as a nonlinear function of the
screen-level air temperature, relative humidity, cloud fraction, total column cloud liquid, and ice
water. A comprehensive evaluation of this new model relative to eight existing models was
conducted under clear-sky and all-sky conditions at daytime/nighttime hourly and daily scales.
The validation results showed that the accuracy of the newly constructed model is superior to
other models, yielding overall RMSE values of 14.82 and 10.76 $W/m^2$ under clear-sky
conditions, and 15.95 and 10.27 $W/m^2$ under all-sky conditions, at hourly and daily scales,
respectively. Our analysis indicates that the effects of the total column cloud liquid and ice water
on the ocean-surface $R_l$ also need to be considered besides cloud cover. Overall, the newly
developed model has strong potential to be widely used.
*Keywords*: Ocean surface, longwave radiation, empirical model, buoy

**1 Introduction**


The downward longwave radiation ($R_l$) at the ocean surface is the thermal infrared (4–
100 μm) radiative flux emitted by the entire atmospheric column over the ocean surface (Yu et
al., 2018).The ocean-surface $R_l$ is among the most important components of the heat flux across
the ocean–atmosphere interface, which, in turn, shapes the climate state of both the atmosphere
and ocean (Caniaux, 2005; Fasullo et al., 2009; Fung et al., 1984). Therefore, an accurate
estimate of the ocean-surface $R_l$ is crucial for studies of air–sea interactions and the climate and
oceanic systems.
Although the ocean-surface $R_l$ is routinely measured at most buoy sites, the available
ocean-surface $R_l$ measurements can not meet the needs of various applications because of the
small number of buoys currently employed (especially moored buoys) and their sparse
distribution across global oceans. Another way to get the $R_l$ at the ocean surface is by using
satellite-based or model reanalysis products. The ocean-surface $R_l$ from satellite-derived
products, such as the International Satellite Cloud Climatology Project (ISCCP) (Rossow &
Zhang, 1995; Young et al., 2018) and Clouds and the Earth's Radiant Energy System Synoptic
Radiative Fluxes and Clouds (CERES/SYN1deg) (Doelling et al., 2013; Rutan et al., 2015) is
usually generated using these satellite data and a radiative transfer model, which simulates the
radiative transfer interactions of light absorption, scattering, and emission through the
atmosphere with the input of given atmospheric parameters. However, radiative transfer models
are not widely used in practice because of their complicacy and the difficulties associated with
collecting all essential inputs. The ocean-surface $R_l$ provided in model reanalysis products, such
as the fifth generation of the European Centre for Medium-Range Weather Forecasts atmospheric
reanalysis of the global climate (ERA5) (Hersbach et al., 2020) and the Modern-Era
Retrospective analysis for Research and Applications, Version 2 (MERRA2) (Gelaro et al.,
2017), is produced by assimilating various observations into an atmospheric model to get the



optimal estimates of the state of the atmosphere and the surface (Gelaro et al., 2017). Previous
studies indicated that $R_l$ estimates from satellite-based products are generally in better agreement
with buoy measurements than those obtained from reanalysis products (Pinker et al., 2014;
Pinker et al., 2018; Thandlam & Rahaman, 2019). However, applications of the ocean-surface $R_l$
from these two kinds of products are limited due to their coarse spatial resolutions (most of them
are coarser than 1º), limited periods (especially satellite-based products), and discrepancies in
accuracy and consistency (Cronin et al., 2019). Hence, many parameterization and empirical
models for estimating ocean-surface $R_l$ that can easily be implemented in practical use have been
established during the past few decades (Bignami et al., 1995; Josey, 2003; Zapadka et al., 2001).
Most of the commonly used $R_l$ estimation models were established using the relationship
between $R_l$ and the relevant meteorological variables (i.e., air temperature, humidity, column
integrated water vapor (IWV), and cloud parameters) or oceanic parameters (i.e., bulk sea
surface temperature), which are usually obtained from in situ measurements or model
simulations (Li & Coimbra, 2019; Li et al., 2017; Paul, 2021). It is known that most $R_l$
estimation models were originally developed for the land surface and were applied to the ocean
surface directly without any alterations by assuming the atmospheric conditions are nearly the
same over ocean and land surfaces (Bignami et al., 1995; Clark et al., 1974; Frouin et al., 1988;
Josey, 2003). However, this assumption increases the uncertainty in $R_l$ estimates because of the
significantly different water vapor profiles over ocean and land surfaces (Bignami et al., 1995). A
few models built specifically for $R_l$ estimation at the ocean surface (Bignami et al., 1995; Josey,
2003; Zapadka et al., 2001) were usually developed using limited observations collected from
buoy sites or cruise ships distributed within a specific region; hence, the robustness of these
models were in doubt when applied globally. For example, Josey (2003) proposed a model for $R_l$
estimation at mid-high latitude seas with a satisfactory validation accuracy, but this new model
performed worse over tropical seas with a tendency to underestimate $R_l$ by up to 10–15 W/m$^2$.
Moreover, most of the existing $R_l$ estimation models only work under clear-sky conditions,
which are especially rare over ocean surfaces. Furthermore, most of these models only derive $R_l$
at instantaneous scales, yet the $R_l$ at the daily scale is more preferred across a range of
applications. Therefore, a new, easily implemented model that can derive accurate and robust $R_l$
estimates at the global ocean surface under all-sky conditions at various temporal scales (e.g.,
instantaneous and daily) is required. More details about the existing $R_l$ estimation models are
given in Section 2.

In addition, according to W Wang and Liang (2009b), the uncertainty of the ocean-surface
$R_l$ estimation should be less than 10 W/m$^2$ for climate diagnostic studies. However, the
performances of the most commonly used $R_l$ estimation models at the global ocean surface were
not thoroughly evaluated in previous studies because of the few available in situ measurements.
Fortunately, being aware of the significance of the energy budget in air–sea interactions
(Centurioni et al., 2019), more and more platforms for radiative measuring have been built across
global ocean surfaces during the past decades, so relatively comprehensive ocean-surface $R_l$
measurements can be collected today, which provide a good opportunity for modeling and
comprehensive evaluations.

Overall, the main goal of this research is to establish a new empirical model for
calculating the all-sky ocean-surface $R_l$ at instantaneous and daily scales based on globally
distributed moored buoy measurements and other ancillary information. A comprehensive
evaluation is conducted on the newly developed model relative to eight commonly used models
for ocean-surface $R_l$ estimation under clear- and all-sky conditions at hourly and daily scales.



The organization of this paper is as follows. A review of the eight commonly used $R_l$ estimation
models is presented in Section 2. Section 3 introduces the data sets used in this research and the
methods, including the new model development and model evaluation. Section 4 shows the
results of the model validation, comparison, and analysis. The key conclusions and discussions
are provided in Section 5.
**2 Review of Previous Models**
Many models were proposed for $R_l$ calculation under various sky conditions at different
temporal scales in previous studies. In this study, eight widely used models were selected for
evaluation and Table 1 shows their basic information. According to the sky conditions under
which these models could be used, the eight $R_l$ estimation models were divided into two classes:
$R_l$ models under clear-sky conditions and under all-sky conditions, respectively. Details of the
eight models are provided one by one in the following section. Note that the downward direction
is defined as positive in this study.
**Table 1**
*Eight Existing Models for Ocean-surface $R_l$ Estimation*

| Sky Condition | Model | Abbr | Designed temporal scale | Reference |
|---|---|---|---|---|
| Clear-sky | $R_l=a\sigma T_a^4(1+b\sqrt{e})$ | Mod1 | Monthly | Brunt (1932) |
| | $R_l=\sigma T_a^4\{1-a\exp(-b(273-T_a)^2)\}$ | Mod2 | 5–15 minute | Idso and Jackson (1969) |
| | $R_l=a\sigma T_a^4(e/T_a)^{1/7}$ | Mod3 | Instantaneous | Brutsaert (1975) |
| | $R_l=a\sigma T_a^4[1-\exp(-e^{T_a/2016})]$ | Mod4 | Daily | Satterlund (1979) |
| | $R_l=\sigma T_a^4[1-(1+\varepsilon)\exp\{-(1.2+3\varepsilon)^{1/2}\}]$ $\varepsilon=46.5(\frac{e}{T_a})$ | Mod5 | Instantaneous | Prata (1996) |
| All-sky | $R_l=\frac{\varepsilon\sigma T_s^4-\varepsilon\sigma T_s^4(a+b\sqrt{e})(1-\lambda C^2)+4\varepsilon\sigma T_s^3(T_s-T_a)}{1-\alpha_1}$ | Mod6 | Daily | Clark et al. (1974) |
| | $R_l=\sigma T_a^4(a+be)(1+dC^2)$ | Mod7 | Hourly | Bignami et al. (1995) |
| | $R_l=\sigma\{T_a+aC^2+bC-d+g(D+f)\}^4$ | Mod8 | Hourly | Josey (2003) |

2.1 Under clear-sky condition
Among the eight models, there are five $R_l$ estimation models that could only be used
under clear-sky conditions.
Brunt (1932) developed the first $R_l$ estimation model (named Mod1) for land surfaces,
which relates the monthly mean $R_l$ to the screen-level water vapor and air temperature, as
Equation (1) shows:

$$R_l=a_1\sigma T_a^4(1+b_1\sqrt{e}) \tag{1}$$

where $a_1$ and $b_1$ are empirical coefficients, $T_a$ is the monthly mean screen-level air





temperature (K), e is the monthly mean screen-level water vapor pressure (mbar), and $\sigma$ is the
Stefan–Boltzmann constant, defined as $5.67 \times 10^{-8} W/(m^2 \cdot K^4)$. In the study of Brunt (1932), the
two coefficients $a_1$ and $b_1$ were suggested as 0.52 and 0.125 based on observations collected from
Benson, South Oxfordshire, England. The validation results of Mod1 showed a correlation
coefficient as high as 0.97 based on the collected samples. However, Swinbank (1963) pointed
out that the validation results of Mod1 for other regions where variations in the humidity and $T_a$
were different from those in Benson were worse. Despite these limitations, as the first empirical
$R_l$ estimation model in a simple format, Mod1 has been widely used to construct the coupling
between hydrological and atmospheric models (Habets et al., 1999; Lohmann et al., 1998).

Different from Mod1, the model developed by Idso and Jackson (1969) (named Mod2)
was based on the theoretical consideration that the effective emittance of an atmosphere is solely
temperature-dependent; hence, the screen-level $T_a$ is the only input of Mod2 for calculating $R_l$:
$$R_l = \sigma T_a^4 \{1 - a_2 \exp(-b_2 (273 - T_a)^2)\} \tag{2}$$

where $a_2$ and $b_2$ are empirical coefficients, which were defined as 0.261 and $7.770 \times 10^{-4}$,
respectively, by Idso and Jackson (1969) based on experimental data at four sites located in
Arizona, Alaska, Australia, and the Indian Ocean, obtained at intervals of 5 to 15 minutes. Idso
and Jackson (1969) thought that Mod2 might be efficient at all latitudes for different seasons, as
it has been developed by using observations from diverse locations. Since publication, Mod2 has
been employed in relevant researches like evaporation estimation (Cleugh et al., 2007; Vertessy
et al., 1993) and ocean-ice modeling (Saucier et al., 2003).

Afterwards, Brutsaert (1975) proposed a simple model for computing $R_l$ by directly
solving the Schwarzschild's transfer equation (Schwarzschild, 1914) under clear skies and
standard atmospheric conditions (i.e., the U.S. 1962 standard atmosphere). This model is denoted
as Mod3, and is described as follows:
$$R_l = a_3 \sigma T_a^4 (e/T_a)^{1/7} \tag{3}$$

where $a_3$ is defined as a constant equal to 1.24, as determined during the Schwarzschild's
transfer equation solving process. Explicit physical theory is reflected in Mod3. The term
$(e/T_a)^{1/7}$, regarded as the atmospheric emissivity, tends to zero when the water vapor content is
very little. However, Prata (1996) indicated that the atmospheric emissivity tends to a certain
constant value even without water vapor, such as values from 0.17 to 0.19 when only $CO_2$ is
present (Staley & Jurica, 1972). The estimates from Mod3 are usually used as the necessary
inputs of hydrological models (Pauwels et al., 2007; Rigon et al., 2006) and climate models
(Mills, 1997).

Aase and Idso (1978) found that Mod2 and Mod3 performed poor when $T_a$ was below
freezing. To address this issue, Satterlund (1979) proposed a model (named Mod4) to compute $R_l$
by reformatting $T_a$ and e, as follows:
$$R_l = a_4 \sigma T_a^4 [1 - \exp(-e^{T_a/2016})] \tag{4}$$

where $a_4$ is an empirical coefficient and defined as 1.08 by Satterlund (1979) based on
collected daily $R_l$ measurements at one site in Sidney, Montana, USA. After validation and
comparison, Satterlund (1979) concluded that Mod4 outperformed Mod2 and Mod3 under
extreme conditions in terms of temperature and humidity and performed comparably with the
two models for other cases. As such, the $R_l$ estimates from Mod4 have been used in studies such





as snow pack evolution (Douville et al., 1995) and hydrological models (Schlosser et al., 1997).
However, because the model does not contain a constant term, the application of Mod4 should be
done with caution if the surface water vapor pressure is very close to zero.
With the development of radiation measuring instruments and technology, several new $R_l$
estimation models have been proposed, such as the model proposed by Prata (1996) (named
Mod5), as follows:
$$R_l = \sigma T_a^4 \left[ 1 - (1 + 46.5(\tfrac{e}{T_a})) \exp \left\{ - \left( a_5 + 46.5 b_5 (\tfrac{e}{T_a}) \right)^{1/2} \right\} \right] \tag{5}$$

where $a_5$ and $b_5$ are empirical coefficients, defined as 1.2 and 3.0 in the study of Prata
(1996) and Robinson (1947; 1950). As with Mod1–Mod4, Mod5 is also dependent on $T_a$ and e
but contains a majorly revised right term (in the square brackets), which is regarded as the
emissivity. After extensive validation and comparison, Prata (1996) claimed Mod5 outperformed
or performed similar to other $R_l$ estimation models, including Mod1–Mod4, in areas within the
polar region, mid-latitudes, and tropical regions. Hence, Mod5 has been applied widely, from
studies of snowmelt modeling (Jost et al., 2009) to urban energy budget (Nice et al., 2018;
Oleson et al., 2008).
To sum up, all five $R_l$ estimation models (Mod1–Mod5) that only work under clear-sky
conditions take $T_a$ and/or e as inputs. Such an approach is in agreement with the research of
Kjaersgaard et al. (2007) who found that $R_l$ is mainly emanated from the low-level atmosphere
that can be adequately characterized in terms of $T_a$ and humidity under clear-sky conditions
(Diak et al., 2000; Ellingson, 1995; Prata, 1996). Moreover, the five models were all established
by using measurements from different regions at various timescales, and they can be employed at
any timescale (see Table 1) regardless of the temporal resolution of the original measurements
used for modeling.
### 2.2 Under all-sky condition
Three $R_l$ estimation models that can work under all-sky conditions were evaluated in this
paper. Comparing to the above five models, ancillary information (e.g., clouds) should be taken
into account in addition to $T_a$ and e in the three models, and the three models were developed
specifically for ocean surfaces.
Based on the model developed by Clark et al. (1974) for the all-sky net longwave
radiation at the ocean surface ($R_{lnet}$, the difference between the downward and upward longwave
radiation) calculation, Josey (2003) proposed a revised model (named Mod6) to estimate the all-
sky ocean-surface $R_l$ by getting rid of the ocean-surface upward longwave radiation as:
$$R_l = \frac{\varepsilon_s \sigma SST^4 - \varepsilon_s \sigma SST^4 (a_6 + b_6 \sqrt{e})(1 - \lambda C^2) - 4\varepsilon_s \sigma SST^3 (SST - T_a)}{1 - \alpha_s} \tag{6}$$

where $\varepsilon_s$ is the sea surface emissivity, defined as a constant value of 0.98, and SST is the
sea surface temperature (K); hence, the term $\varepsilon_s \sigma SST^4$ is the upward longwave radiation at the
ocean surface. $\alpha_s$ is the sea surface longwave radiation reflectivity, defined as a constant value of
0.045, C is the cloud cover (0–1; dimensionless), $\lambda$ is a latitude-dependent coefficient that
represents the cloud amount, and $a_6$ and $b_6$ are empirical coefficients. Based on measurements
(i.e., $R_l$, $T_s$, and C) collected from the Chemical and Hydrographic Atlantic Ocean Section
(CHAOS) in the northeast Atlantic in 1998, $a_6$ and $b_6$ were determined as 0.39 and -0.05 (Clark et
al., 1974; Josey, 2003), and λ at a given latitude can be taken from Josey et al. (1997). Josey
(2003) validated Mod6 and the results showed that Mod6 tended to overestimate the
instantaneous $R_l$ measurements from CHAOS by 11.70 W/m$^2$. The estimates from Mod6 have
been applied in hydrodynamic models (Grayek et al., 2011) and atmospheric boundary layer
models (Deremble et al., 2013).
Based on hourly cruise measurements (i.e., $R_l$, $T_a$, and C) collected in the Mediterranean
Sea during the period from 1989 to 1992, Bignami et al. (1995) proposed an empirical model to
calculate the ocean-surface all-sky $R_l$ (named Mod7) as follows:
$$R_l=\sigma T_a^4(a_7+b_7e)(1+c_7C^2) \tag{7}$$
where $a_7$, $b_7$, and $c_7$ are empirical coefficients defined as 0.684, 0.0056, and 0.1762,
respectively. Bignami et al. (1995) presented validated RMSE values for Mod7 which ranged
from ~14 W/m$^2$ at the hourly scale to ~9 W/m$^2$ at the daily scale. Mod7 has been utilized by the
Mediterranean Forecasting System for predictions of currents and biochemical parameters
(Pinardi et al., 2003), coupled ocean–atmosphere climate models (Dubois et al., 2012) as well as
generation of the Atlantic Ocean heat flux climatology (Lindau, 2012).
Also based on the measurements collected from CHAOS, Josey (2003) assessed the
accuracy of Mod7 and found that this model tended to underestimate the all-sky $R_l$ by 12.10
W/m$^2$ at the instantaneous scale. After analyzing the shortcomings of Mod6 and Mod7, Josey
(2003) proposed a new model (named Mod8) for all-sky ocean-surface $R_l$ calculation through a
revision of $T_a$ by using the same samples:
$$R_l=\sigma\{T_a+a_8C^2+b_8C-c_8+d_1(D+e_1)\}^4 \tag{8}$$
where $a_8$, $b_8$, $c_8$, $d_1$, and $e_1$ are empirical coefficients determined as 10.77, 2.34, 18.44,
0.84, and 4.01, respectively, D is the dew point depression, and $T_a$ is the temperature (K) (see
Equation (11)). Estimates of $R_l$ obtained with Mod8 agreed to within 2 W/m$^2$ in the mean bias of
10 minute measurements at middle-high latitudes. The estimates from Mod8 have been used as
essential input in simulations of ocean–atmosphere interactions in the Arctic shelf (Cottier et al.,
2007).

Overall, it was thought that variations in the all-sky ocean-surface $R_l$ were related to $T_a$, e,
and cloud information (e.g., cloud cover and cloud amount) in previous studies. However, Fung
et al. (1984) pointed out that other relevant cloud information, such as the cloud base height
(CBH) and cloud optical thickness, also have a significant influence on ocean-surface longwave
radiation. Therefore, more efforts should be made to increase the $R_l$ estimation accuracy under
all-sky conditions.

## 3 Data and Methodology

In order to develop a new all-sky ocean-surface $R_l$ estimation model, the meteorological
and radiative observations from 65 moored buoys and the cloud parameters from the ERA5
reanalysis product from 1988 to 2019 were applied. Afterwards, the newly developed model and
the eight commonly used models (Mod1–Mod8) were evaluated against the moored $R_l$
measurements under clear- and all-sky conditions at hourly/daily scales

3.1 Data and pre-processing

Table 2 lists all the variables employed in this paper and their information. The

instantaneous timescale can be defined as timescales ranging from a 3 minute average to hourly
average (Bignami et al. (1995); K Wang and Liang (2009a); hence, two timescales, hourly and
daily, were considered in this study for model evaluation as in previous studies (Bilbao & de
Miguel, 2007; Kjaersgaard et al., 2007; Sridhar & Elliott, 2002). Note that Mod1 was also used
at the two timescales (Guo et al., 2019) though it was originally established with monthly
samples. More details about the data are given below.
**Table 2**
*Variables: Explanations and Sources*

| Abbreviation | Full name | Time scales | Unit | Source |
|---|---|---|---|---|
| RH | Relative humidity | Daily/hourly | % | In situ |
| e | Water vapor | Daily/hourly | hPa | Calculated |
| $T_a$ | 2-m air temperature | Daily/hourly | K | In situ |
| $T_s$ | Sea surface temperature | Daily/hourly | K | In situ |
| D | Dew point depression | Daily/hourly | K | Calculated |
| CI | Clearness index | Daily/hourly | 0-1 | Calculated |
| C | Fractional cloud cover | Daily/hourly | 0-1 | Calculated |
| clw | Total column cloud liquid water | Daily/hourly | $g/m^2$ | ERA5 |
| ciw | Total column cloud ice water | Daily/hourly | $g/m^2$ | ERA5 |

3.1.1 Measurements from moored buoys

All measurements were collected from 65 moored buoy sites, whose latitudes range from

47°S to 59.5°N, as shown in Figure 1. The majority of moored buoy sites were located in
tropical seas (23.5°S–23.5°N), and relatively few buoys were in the high-latitude seas of the
Northern Hemisphere (>50°N) and the mid-high latitude seas of the Southern Hemisphere
(>30°S).

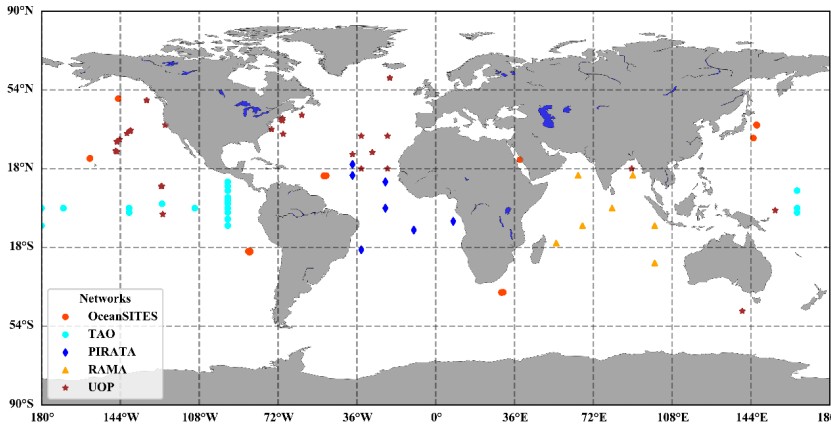


**Figure 1.** Spatial distribution of the 65 moored buoys.

The moored buoy sites in this study belong to five well-known observation

network/programs, including the Upper Ocean Processes Group (UOP), Tropical Atmosphere
Ocean/Triangle Trans-Ocean Buoy Network (TAO/TRITON), Pilot Research Moored Array in





the Tropical Atlantic (PIRATA), Research Moored Array for African–Asian–Australian Monsoon
Analysis and Prediction (RAMA), and OceanSITES. Launched by the Woods Hole
Oceanographic Institution (WHOI), UOP mainly focuses on studying the physical processes of
the air-sea interface and the epipelagic, and its buoys are equipped with oceanographic and
meteorological sensors. The UOP measurements accurately quantify annual cycles of wind stress
and net air-sea heat exchange in the Southern Ocean (Schulz et al., 2012). Twenty-two sites form
the UOP, and data from all were used in this study. TAO/TRITON (McPhaden et al., 1998) in the
tropical Pacific, PIRATA (Bourlès et al., 2008) in the tropical Atlantic, and RAMA in the tropical
Indian Ocean (McPhaden et al., 2009) are all part of the Global Tropical Moored Buoy Array
(GTMBA) program (McPhaden et al., 2010). Extensive quality control was done by GTMBA
prior to dissemination of the data (Freitag, 1999; 2001; Lake, 2003; Medovaya et al., 2002), and
they have been used for monitoring, understanding, and forecasting the El Niño–Southern
Oscillation (ENSO) and monsoon variability (McPhaden et al., 2009). Data from 35 GTMBA
sites (TAO, 21; PIRATA, 7; RAMA, 7) were used in this study. The OceanSITES network is
composed of buoys funded by oceanographic researchers across the globe. The goal of the
OceanSITES program is to facilitate the use of high-quality multidisciplinary data from fixed
sites in the open ocean (Cronin et al., 2019). Eight sites from OceanSITES were utilized. In this
study, the routine measurements made at moored buoys, including radiative measurements (e.g.,
ocean-surface downward shortwave radiation $R_g$) and meteorological measurements (e.g., $T_a$ and
RH) were collected and used; other variables (e.g., e, D, and CI) were calculated from these
measurements. More information regarding these data sets is found in Table 3.
**Table 3**
*Descriptions of Different Networks*

| Network/Program | No. of sites | Period | Observation frequency | Variables | URL |
|---|---|---|---|---|---|
| UOP | 22 | 1988-2017 | 1 hour | $R_l$, $R_g$, $T_a$,RH | http://uop.whoi.edu/index.html |
| TAO/TRITON | 21 | 2000-2019 | 10 min | $R_l$, $R_g$, $T_a$,RH | https://www.pmel.noaa.gov/tao/drupal/disdel/ |
| RAMA | 7 | 2004-2019 | 10 min | $R_l$, $R_g$, $T_a$,RH | https://www.pmel.noaa.gov/tao/drupal/disdel/ |
| PIRATA | 7 | 2006-2019 | 10 min | $R_l$, $R_g$, $T_a$,RH | https://www.pmel.noaa.gov/tao/drupal/disdel/ |
| OceanSITES | 8 | 2000-2018 | 1 hour | $R_l$, $R_g$, $T_a$,RH | http://www.oceansites.org/ |

3.1.1.1 Radiative measurements
At each moored buoy, $R_l$ is routinely measured by an Eppley Precision Infrared
Radiometer (PIR) with a nominal accuracy of ±1% (Richard E. Payne & Anderson, 1999), and
$R_g$ is routinely measured by an Eppley Laboratory precision spectral pyranometer (PSP) with a
calibration accuracy of ±2% (Freitag, 1994). The PIR and PSP are deployed approximately 3 m
above sea level. All measurements are quality controlled by their providers. To ensure data
quality, a two step approach was implemented; 1) only observations flagged as 'high quality' by
the data providers were considered, and 2) data was manually inspected by the authors for any
irregularities. Additionally, the $R_l$ measurements above 450 W/m$^2$ were removed, as suggested
by Josey (2003).
As pointed out by Pascal and Josey (2000), the main errors in measuring $R_l$ are from the



shortwave leakage and differential heating of the sensor. Therefore, the errors ($\Delta R_l$) in $R_l$
observations were corrected according to Pascal and Josey (2000) as:

$$\Delta R_l = (a+\lambda)R_g + bR_g^2 \qquad (9)$$

where $a = 4.34 \times 10^{-3}$, $\lambda = 0.011$, and $b = 1.72 \times 10^{-6}$. Hence, the $R_l$ measurements at a
sampling frequency less than one hour were first corrected. After that, selected measurements
whose sampling frequency was less than one hour were aggregated into hourly means as long as
80% of the measurements in one hour were available, and the hourly data were aggregated into
daily means as long as 24 hourly data in one day were available.
Note that the errors of the measured $R_g$ induced by buoy rocking motions, sensor tilting,
and aerosol accumulation (Medovaya et al., 2002) were too small to be considered here. At last,
47,266 samples at the daily scale and 1,275,308 samples at the hourly scale during the period
from 1988 to 2019 were used in this study. For better comparison, the hourly samples used for
independent validation were further divided into daytime ($R_g > 120$ W/m$^2$) and nighttime
conditions ($R_g \leq 120$ W/m$^2$), with 147,981 samples in daytime and 210,057 in nighttime.

### 317  3.1.1.2 Meteorological and oceanic variables

Two meteorological measurements, RH and $T_a$, were collected at the moored buoy sites.
The instrument used for measuring RH and $T_a$ is a Rotronic MP-100F, deployed about 3 m above
the sea level. The instrument produced accuracies of 2.7% and 0.2 K (Lake, 2003) for RH and
$T_a$, respectively, which are also too small to influence the accuracy of the $R_l$ estimation. Similar
to the radiative measurements, RH and $T_a$ were both strictly screened and then aggregated into
hourly and daily means.
On the contrary, the sea surface temperature (SST) was measured at about 1 m below the
sea level using a high-accuracy conductivity and temperature recorder (SBE37/39; Sea Bird
Electronics) with an accuracy of 0.002 K. According to Donlon et al. (2002), there is a strong
correlation between body SST and skin SST. Although wind speed has a significant effect on this
relationship, a constant correction offset can be applied when the wind speed exceeds 6 m/s
(Alappattu et al., 2017). In fact, 83% of the samples had wind speeds above 4 m/s, and as
suggested by Vanhellemont (2020), the bulk SST measured at moored buoys can be adjusted to
the skin SST by using a correction offset of 0.17 K.

### 332  3.1.1.3 *Calculation of other variables*

Three variables, including e, D, and CI, were calculated with the RH, $T_a$, and $R_g$,
measurements separately. Therefore, these three variables at hourly and daily scales were
obtained from the corresponding measurements. Specifically, the daily (hourly) mean e was
calculated from the daily (hourly) RH using the following equation:

$$e = 6.1121 \frac{RH}{100} \exp\left(\frac{17.502 T_a}{T_a + 240.97}\right) \qquad (10)$$

Note that Equation (10) only works when $T_a$ is in the range -30–50 °C (Buck, 1981), and
$T_a$ should be in items of °C.
The daily (hourly) dew point depression D was calculated according to Josey (2003) and
Henderson-Sellers (1984) as:





$$D = 34.07 + 4157/\ln(2.1718*10^8/e) - T_a \qquad (11)$$

The clearness index (CI) is calculated as the ratio of the surface $R_g$ to the extraterrestrial
solar radiation ($DSR_{toa}$) (Ogunjobi & Kim, 2004). CI generally represents the atmospheric
transmissivity affected by permanent gases, aerosols, and the optical thickness of the clouds
(Alados et al., 2012; Flerchinger et al., 2009; Gubler et al., 2012; Jiang et al., 2015; Meyers &
Dale, 1983), and it is widely used in radiation related researches (Iziomon et al., 2003; Jiang et
al., 2016; Jiang et al., 2015; Richard E Payne, 1972). The value of CI is between 0 and 1, where a
larger CI value represents a clearer sky. The hourly CI can be calculated as follows:
$$CI = \frac{R_g}{DSR_{toa}} \qquad (12)$$

However, during nighttime, the hourly CI cannot be calculated by Equation (12) directly
because of a lack of $R_g$ values; hence, it was calculated based on a 24-hour solar radiation
window centered on the hourly observation as suggested by Flerchinger et al. (2009). The daily
CI was calculated as the average of all hourly CI values in a day for the sake of considering
atmosphere variations at nighttime.
In this paper, CI was utilized to determine the condition as clear-sky when its value was
greater than 0.7 at both hourly and daily scales. Additionally, it was found that the cloud cover
derived from CI would help to improve the model performance after multiple experiments,
especially at nighttime. Therefore, CI was also used to calculate the cloud cover. Specifically, the
cloud fraction was linearly interpolated between C = 1.0 at a CI value of 0.4 for complete cloud
cover to C = 0.0 at a CI value of 0.7 for cloudless, both at daily and hourly scales according to
Flerchinger et al. (2009). Because of the different calculation of CI during daytime and
nighttime, the uncertainty in the calculated cloud cover was different; hence, the $R_l$ estimates at
the hourly scale were further examined at daytime and nighttime. Therefore, all meteorological
factors (RH, $T_a$, e, and D) at daily and at hourly scales were respectively prepared accordingly.
3.1.2 Cloud parameters from the ERA5 reanalysis data set
As described above, the cloud cover represented by the fraction (C) is usually taken into
account when estimating $R_l$ affected by clouds. However, in this study, two more cloud-related
parameters, including clw and ciw (see Table 1), from the ERA5 reanalysis product were also
considered in the modeling. The total amount of liquid water per unit area in the air column from
the base to the top of the cloud is called the total column cloud liquid water (clw), and its chilled
counterpart (ice) is called the total column cloud ice water (ciw) (Nandan et al., 2022). ERA5 is
the fifth generation atmospheric reanalysis product, and it was produced based on 4D-Var data
assimilation using the Integrated Forecasting System (IFS) with an enhanced spatial resolution
(0.25°) and time resolution (hourly) compared to its previous version ERA-interim (Hoffmann et
al., 2019) from 1979 to present. Clouds in ERA5 are represented by a fully prognostic cloud
scheme, in which cloud fractions and cloud condensates obey mass balance equations (Tiedtke,
1993). The ERA5 clw values are in good agreement with those obtained from radiosonde
observations (Nandan et al., 2022). Overall, relative to ERA-interim, ERA5 shows reduced
biases in the total ice water path versus other satellite-based observational products. Therefore,
the two cloud parameters were extracted from the locations of the 65 moored buoy sites directly
at the hourly scale, and then their daily means were calculated by averaging the 24 valid hourly
values. ERA5 cloud product is available on the Climate Data Store (CDS) cloud server
(https://cds.climate.copernicus.eu/cdsapp#!/search?type=dataset).





Overall, 70% of the samples at each moored buoy site, including 33,151 daily samples
and 917,270 hourly samples, were randomly selected for new model training and calibration of
the eight previous models (Mod1– Mod8). The other 30% of the data at each site, including
14,115 daily samples and 358,038 hourly samples (daytime: 147,981; nighttime: 210,057), were
used for model validation.
### 3.2. Methodology
A new model that could estimate ocean-surface $R_l$ under all-sky conditions at both hourly
and daily scales was developed based on the moored measurements and ERA5 cloud parameters.
Moreover, the eight evaluated $R_l$ models were all recalibrated so as to evaluate the model's
accuracy objectively. Based on the corresponding validation samples, the $R_l$ values produced by
the nine models were compared under clear-sky and all-sky conditions at hourly and daily scales,
where the comparison at the hourly scale was further divided into daytime and nighttime values.
#### 3.2.1 New $R_l$ estimation model development
As mentioned above, $T_a$ and the humidity-related factors (e.g., RH) were enough to
characterize the variations in $R_l$ under clear-sky conditions. However, for cloudy skies, $R_l$ is
enhanced by the cloud base emitting (T Wang et al., 2020; Yang & Cheng, 2020). Cloud cover is
one of the most commonly used cloud-related parameters. In addition, theoretically, the cloudy-
sky $R_l$ is significantly influenced by the cloud's base temperature, which is determined by the
CBH; hence, CBH is thought to be necessary in determining $R_l$ under cloudy-sky conditions
(Viúdez-Mora et al., 2015). However, it is difficult to obtain the CBH accurately, especially for
partly cloudy skies (Zhou & Cess, 2001) because of the unavailability of the cloud's geometrical
thickness (Yang & Cheng, 2020). Therefore, other parameters that could provide information on
the CBH were explored. In the study of Hack (1998), a physical correlation between clw and
CBH was revealed for most cases, while clw was successfully used as an effective surrogate of
the CBH in the study of Zhou and Cess (2001). However, Zhou et al. (2007) pointed out that the
effects of ice clouds on $R_l$ should also be considered when the atmospheric water vapor is low or
at high latitudes, which means that ciw also needs to be taken into account. Inspired by these
studies, clw and ciw, both in logarithmic form, were introduced in the development of a new
model named Modnew, in which $R_l$ under all-sky conditions at the ocean surface was related to
five parameters including $T_a$, RH, clw, ciw, and C. Modnew was trained by the corresponding
training samples at hourly and daily scales. Details of the development of the new model
presented in the present study are given in Section 4.1.
#### 3.2.2 Model performances evaluation
Table 4 lists the different cases for the $R_l$ model comparison. As shown in Table 4, the
nine evaluated models (Mod1–Mod8 and Modnew) were all used for clear-sky $R_l$ estimation at
both hourly and daily scales, while only four models (Mod6–Mod8 and Modnew) were evaluated
under all-sky conditions. Three metrics were employed to present the model accuracy: $R^2$, the
root-mean-square error (RMSE), and bias. Generally, all three statistics were calculated to
evaluate the accuracy of different models, but the RMSE values had larger weights.
**Table 4**
*Detailed Information of the Six Cases Considered in the Model Evaluation*

| Case | Training | Validation | Evaluated model |
| --- | --- | --- | --- |





| | | | samples | samples | |
|---|---|---|---|---|---|
| Clear-sky | Hourly | Daytime | 176,510 | 40,805 | Mod1-Mod8, Modnew |
| | | Nighttime | | 35,125 | Mod1-Mod8, Modnew |
| | Daily | | 3,443 | 1,447 | Mod1-Mod8, Modnew |
| All-sky | Hourly | Daytime | 917,270 | 147,981 | Mod6-8, Modnew |
| | | Nighttime | | 210,057 | Mod6-8, Modnew |
| | Daily | | 33,151 | 14,115 | Mod6-8, Modnew |

**4 Results and Analysis**
In this section, Modnew is introduced first, and then the validation results of the nine
evaluated models under various cases are compared and analyzed. Lastly, further analyses are
conducted on Modnew.
4.1 Modnew development
As mentioned above, the ocean-surface $R_l$ in Modnew is related to five parameters ($T_a$,
clw, RH, C, and ciw) for hourly and daily scales under all-sky conditions. To understand better
the contribution made by each variable on $R_l$, the five parameters were introduced into Modnew
gradually. Taking the daily all-sky $R_l$ as an example, $R_l$ was first only characterized by the fourth
power of $T_a$ based on the Stefan–Boltzmann law as follows:

$$R_l = a_{new}\sigma T_a^4 + b_{new} \tag{13}$$

where $a_{new}$ and $b_{new}$ are empirical coefficients, determined as 0.85 and 14.96, respectively,
based on the daily training samples. Then, the correlations between the model residuals in $R_l$
(referred to as $\Delta R_l$) that define the difference between the in situ $R_l$ measurements and the $R_l$
estimates from Equation (13) and other four parameters (clw, RH, C, and ciw) were explored one
by one. The results are found in Figure 2.



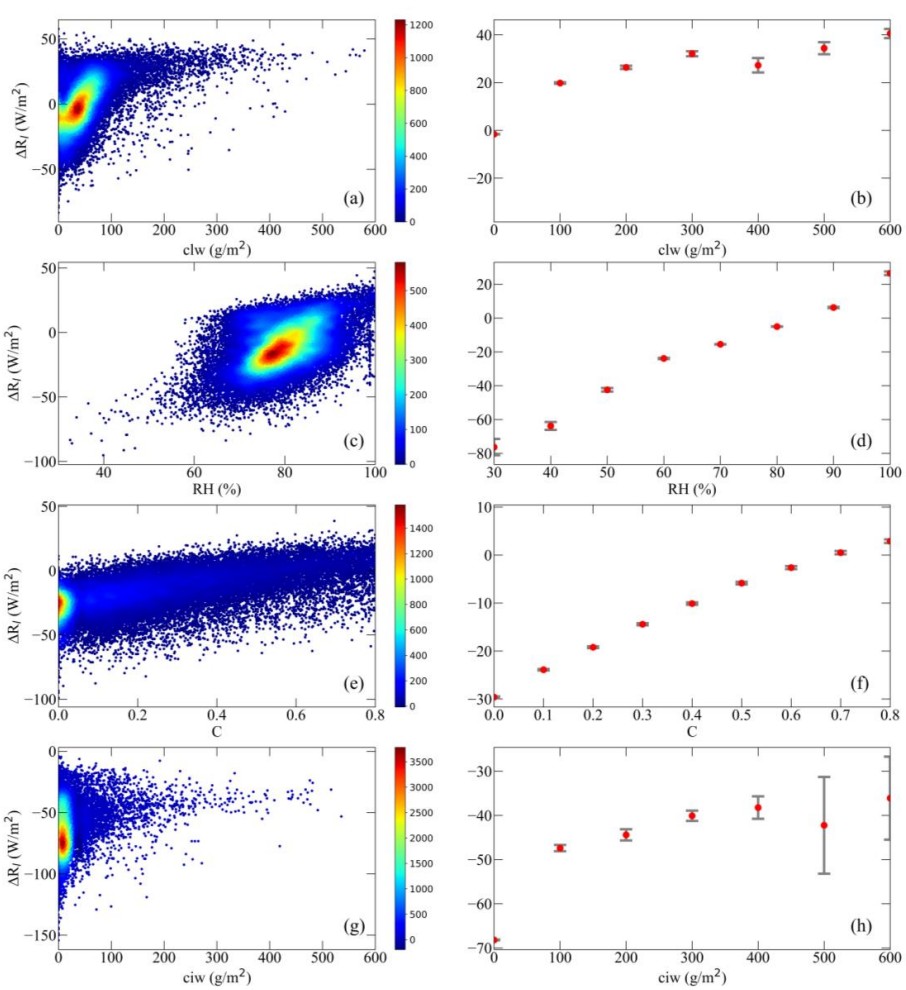


**Figure 2.** The scatter plots between the model residuals, $\Delta R_l$, from Equation (13) and (a) clw, (c)
RH, (e) C, and (g) ciw. Panels (b), (d), (f), and (h) are their corresponding box plots.

Figures 2(a), 2(c), 2(e), and 2(g) present scatter plots between $\Delta R_l$ and clw, RH, C, and
ciw, respectively. In order to show their relationships better, the corresponding box plots, in
which the mean of $\Delta R_l$ and its standard error (SEM) for each bin of the four parameters (in 10%
increments) were calculated and presented in Figures 2(b), 2(d), 2(f), and 2(h), respectively.
Specifically, $\Delta R_l$ varied with clw and ciw in a logarithmic relationship (Figures 2(b) and 2(h),
respectively), and with RH (Figure 2(d)) and C (Figure 2(f)) in approximately linear
relationships. Wefound that by introducing the C, RH, clw and ciw in Equation (13) gradually,
the RMSE error was reduced from 17.48 W/m² with Equation (13) to 12.61 W/m², 10.92 W/m²,
10.11 W/m² and 9.87 W/m², and the level of $R^2$ increased accordingly from 0.64 to 0.81, 0.86,
0.88 and 0.89, respectively. Hence, clw, RH, C, and ciw were introduced into Equation (13) in
their appropriate forms and the final equation was taken as Modnew:

$$R_l = a_{new}\sigma T_a^4 + b_{new}C + c_{new}\ln(1 + clw) + d_{new}\ln(1 + ciw) + e_{new}RH + f_{new}$$



(14)

where $a_{new}$, $b_{new}$, $c_{new}$, $d_{new}$, $e_{new}$, and $f_{new}$ are empirical coefficients. In this study, these
coefficients were determined as 1.06, 42.18, 4.90, -1.97, 0.89, and -178.28 respectively. Figure
3(a) shows that the overall training accuracy of the estimated all-sky ocean-surface $R_l$ from
Modnew was satisfactory, yielding an $R^2$ of 0.89, RMSE of 9.87 W/m$^2$, and nearly no bias.
Afterwards, Equation (14) was used to determine the hourly ocean-surface $R_l$ based on the
corresponding hourly training samples (see Table 4). The hourly results shown in Figure 3(b)
were satisfactory, with an $R^2$ of 0.78, RMSE of 15.44 W/m$^2$, and nearly no bias. Note that the $R_l$
measurements whose values were larger than 450 W/m$^2$ were thought to be unreasonable and
were manually removed (see Section 3.1).

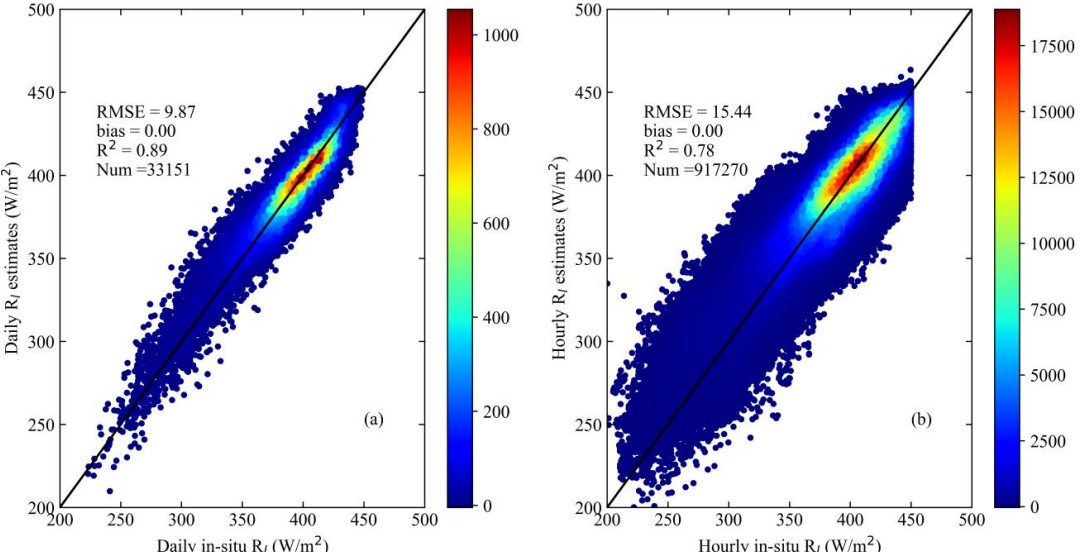


**Figure 3.** Overall training accuracy of the all-sky daily $R_l$ at (a) daily and (b) hourly scales.

By considering the influence of the calculated cloud cover on the $R_l$ estimates, the hourly
results were separated into daytime and nighttime, respectively, as shown in Figure 4. The
training accuracy of the daytime sample was higher than that at nighttime, with $R^2$ values of 0.82
and 0.79 and RMSE values of 13.18 and 16.24 W/m$^2$, respectively. It was assumed that the larger
uncertainties in the hourly ocean-surface $R_l$ at nighttime were possibly owing to the estimated
cloud cover, which might have an influence on Modnew in the form of overestimating $R_l$.
Overall, the performance of Modnew was very good, both at daily and hourly scales for all-sky
$R_l$ estimation at the ocean surface.



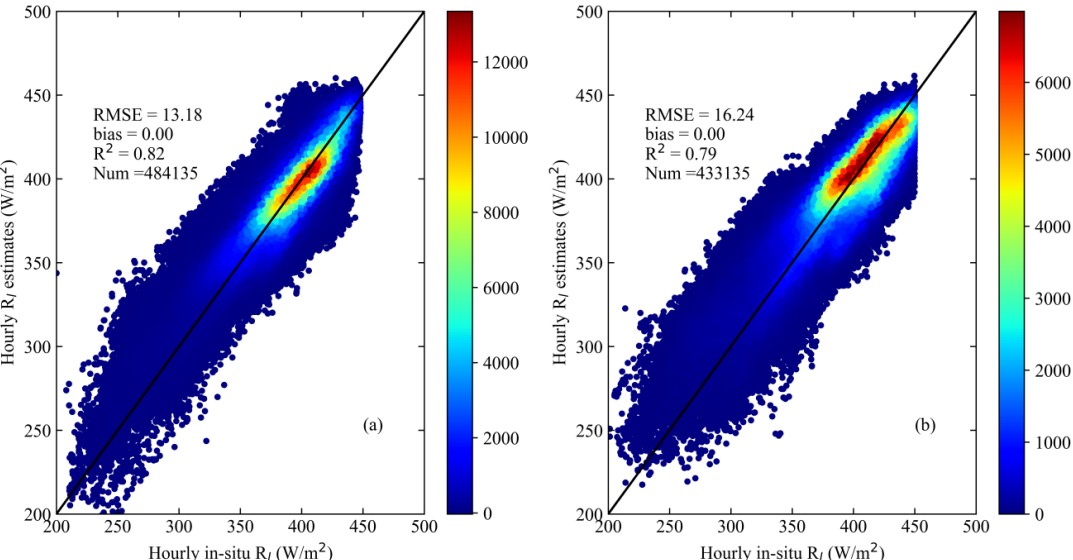

**Figure 4.** Overall training accuracy of the all-sky hourly $R_l$ during (a) daytime and (b) nighttime.

### 4.2 Model comparison results

Based on the independent validation samples, Mod1–Mod8 and Modnew were validated one by one and compared for various cases (Table 4). Before that, the eight existing models were calibrated using the corresponding training samples, which means that Mod1–Mod5 were calibrated with the clear-sky training hourly/daily samples, while Mod6–Mod8 were calibrated with the all-sky training hourly/daily samples, i.e., the same as Modnew. Afterwards, these models were validated against the matched validation samples for each case. The updated coefficients of Mod1–Mod8 and the coefficients of Modnew for hourly and daily scales are given in Table 5. For better illustration, the comparison results are presented for clear- and all-sky conditions in the following paragraphs.

**Table 5**

*Coefficients of the Nine Models Used for Hourly/Daily Ocean-surface $R_l$ Estimation. The Values in Parentheses are the Uncertainties of the Fitted Parameters*

| Models | a | b | c | d | e | f |
|---|---|---|---|---|---|---|
| ***Hourly*** | | | | | | |
| Mod1 | $0.675(\pm 6 \times 10^{-4})$ | $0.052(\pm 3 \times 10^{-4})$ | / | / | / | / |
| Mod2 | $0.246(\pm 1 \times 10^{-4})$ | $7.77 \times 10^{-4}(\pm 0.03)$ | / | / | / | / |
| Mod3 | $1.21(\pm 9 \times 10^{-5})$ | / | / | / | / | / |
| Mod4 | $1.056(\pm 8 \times 10^{-5})$ | / | / | / | / | / |
| Mod5 | $7.48(\pm 0.01)$ | $1.28(\pm 0.003)$ | $0.5(\pm 0.005)$ | / | / | / |
| Mod6 | $0.229(\pm 4 \times 10^{-4})$ | $-0.006(\pm 8 \times 10^{-5})$ | / | / | / | / |
| Mod7 | $0.812(\pm 2 \times 10^{-4})$ | $0.001(\pm 7 \times 10^{-6})$ | $0.121(\pm 1 \times 10^{-4})$ | / | / | / |
| Mod8 | $-5.557(\pm 0.38)$ | $13.378(\pm 0.35)$ | $82.43(\pm 1.21)$ | $0.85(\pm 0.02)$ | $85.33(\pm 0.60)$ | / |
| Modnew | $0.986(\pm 6 \times 10^{-4})$ | $40.991(\pm 0.05)$ | $3.116(\pm 0.01)$ | $-2.478(\pm 0.01)$ | $0.921(\pm 0.02)$ | $-144.62(\pm 0.30)$ |





| *Daily* | | | | | | |
|---|---|---|---|---|---|---|
| Mod1 | 0.65(±0.004) | 0.06(±0.001) | / | / | / | / |
| Mod2 | 0.25(±0.003) | 7.77×$10^{-4}$(±0.18) | / | / | / | / |
| Mod3 | 1.21(±5 ×$10^{-4}$) | | / | / | / | / |
| Mod4 | 1.061(±5 ×$10^{-4}$) | / | / | / | / | / |
| Mod5 | 1.69(±0.09) | 2.67(±0.25) | 0.5(±0.02) | / | / | / |
| Mod6 | 0.286(±0.002) | -0.03(±3 ×$10^{-4}$) | / | / | / | / |
| Mod7 | 0.805(±0.002) | 0.002(±8 ×$10^{-5}$) | 0.133(±0.01) | / | / | / |
| Mod8 | -0.34(±0.02) | 8.545(±0.19) | -12.19(±0.59) | 0.08(±0.009) | 0.08(±0.006) | / |
| Modnew | 1.06(±0.002) | 42.18(±0.22) | 4.90(±0.06) | -1.97(±0.04) | 0.89(±0.008) | -178.28(±1.15) |

### 4.2.1 Clear sky

All models, including the eight previous models (Mod1–Mod8), and the newly developed model (Modnew), could be used under clear-sky conditions at both hourly and daily scales with the updated coefficients given in Table 5.

### 4.2.1.1 Hourly scale

Table 6 shows the validation results of the nine models under clear-sky conditions at the hourly scale. Meanwhile, the validation results of Mod1–Mod8 with their original coefficients (see Section 2) are also presented in Table 6, using the same validation samples for comparison.

**Table 6**

*Overall Validation Accuracy of the Nine Ocean-surface $R_l$ Models under Clear-sky Conditions at the Hourly Scale. The Values in Parentheses for Mod1–Mod8 are the Validation Results Found Using Their Original Coefficients*

| Models | $R^2$ | RMSE(W/m$^2$) | bias(W/m$^2$) |
|---|---|---|---|
| Mod1 | 0.77 (0.78) | 14.69 (15.43) | -0.42 (-0.88) |
| Mod2 | 0.71 (0.71) | 16.37 (16.61) | -0.31 (-2.80) |
| Mod3 | 0.77 (0.77) | 14.77 (17.87) | -0.53 (9.84) |
| Mod4 | 0.74 (0.74) | 15.53 (17.11) | -0.22 (7.33) |
| Mod5 | 0.77 (0.77) | 14.62 (26.90) | -0.44 (-19.56) |
| Mod6 | 0.75 (0.77) | 16.87 (21.51) | 7.33 (15.28) |
| Mod7 | 0.74 (0.77) | 18.37 (17.52) | 9.27 (-9.57) |
| Mod8 | 0.78 (0.78) | 15.59 (37.00) | 2.45 (-33.27) |
| Modnew | 0.79 | 14.82 | 4.40 |

The validation results illustrate that most models estimated the clear-sky hourly ocean-surface $R_l$ with a similar accuracy, with $R^2$ values ranging from 0.74 to 0.79, RMSE values ranging from 14.62 to 18.37 W/m$^2$, and bias values ranging from -0.53 to 9.27 W/m$^2$ (Table 6). All eight existing models with the calibrated coefficients had a higher accuracy than those with the original coefficients except Mod7; in particular, the RMSE of Mod8 decreased by ~21 W/m$^2$. The magnitude of the bias of Mod1–Mod8 also decreased after recalibration, with the magnitudes of the biases of Mod1–Mod5 being much smaller than those of Mod6–Mod8 and Modnew, which were trained with the all-sky hourly samples. Among the four all-sky models, the newly developed Modnew performed the best, with the largest $R^2$ of 0.79, the smallest RMSE of 14.82 W/m$^2$.

Then, the hourly validation results of the nine models were further examined using the





daytime and nighttime values separately, which are shown in Figure 5. The performance of most models, including the five clear-sky models (Mod1–Mod5) and one all-sky model (Mod8), in estimating the hourly clear-sky $R_l$ during the daytime was much better than that at nighttime, with RMSE values at daytime and nighttime ranging from ~12.50 to 15.06 W/m$^2$ and 16.80 to 19.50 W/m$^2$, respectively. On the contrary, the performances of Mod6–Mod7 and Modnew were better at nighttime than that at daytime, with RMSE values at daytime and nighttime ranging from ~15.00 to 19.20 W/m$^2$ and 14.40 to 16.60 W/m$^2$, respectively. Regarding the bias values, at nighttime, all five clear-sky models had a significant underestimation problem (negative biases), while the all-sky models had smaller bias values. This may be due to the uncertainty in the calculated CI at nighttime, which could influence the cloud determination and then $R_l$. In addition, among the five clear-sky models, Mod2 based only on air temperature shows the lowest accuracy in terms of RMSE during both daytime and nighttime. Among the nine models, Modnew had the most stable performance in hourly $R_l$ estimation under clear-sky conditions during both daytime and nighttime with similar RMSE values of 15.03 and 14.38 W/m$^2$, respectively, where in particular its nighttime $R_l$ estimation accuracy was the best among the nine models.

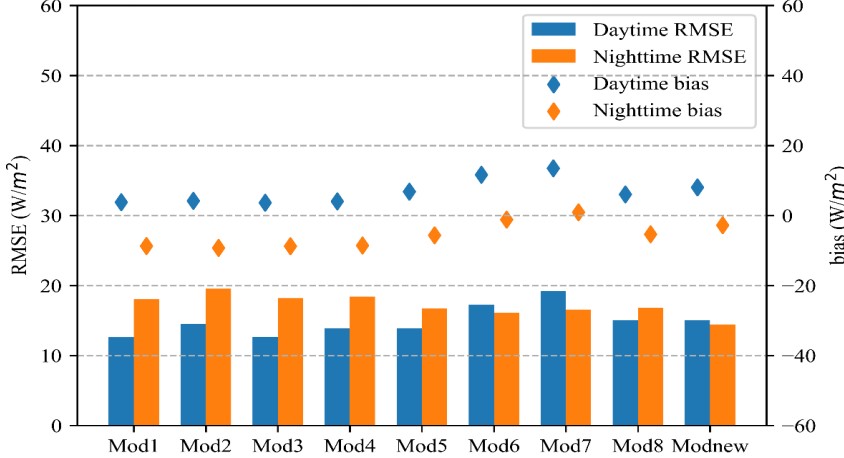

**Figure 5.** Validation accuracy of the estimated $R_l$ under clear-sky conditions at the hourly scale for the nine models represented by RMSE (left axis) and bias (right axis).

Furthermore, the four all-sky $R_l$ estimation models (Mod6–Mod8 and Modnew) were also trained using the clear-sky hourly samples, and their outputs were validated against the in situ observations. The estimation accuracy of the four all-sky models all improved after calibration: their overall validated RMSE values decreased to ~13.40 to 15.40 W/m$^2$ and ~12.01 to 14.29 W/m$^2$ during the daytime, slight decreases (~1 W/m$^2$) at nighttime, and their biases values tended to 0. This indicates that the ability of the four all-sky models in estimating clear-sky hourly $R_l$ was comparable with or even better than the other five models which only work for clear-sky conditions. Indeed, Modnew performed the best of all models during either daytime or nighttime, with corresponding validated RMSE values of 12.01 and 16.00 W/m$^2$, respectively.

4.2.1.2 Daily scale

As for the results at the daily scale, the nine evaluated models were trained with the





corresponding daily training samples (see Table 4) and validated against the in situ
measurements. As shown in Table 7, the estimation accuracy of the daily clear-sky ocean-surface
$R_l$ from nearly all previous models improved significantly after recalibration, where the RMSE
values and the magnitudes of the bias decreased by up to ~4 W/m$^2$ and ~9 W/m$^2$, respectively,
except for Mod7. The five clear-sky models (Mod1–Mod5) performed much better than the three
previous all-sky models (Mod6–Mod8), with RMSE values ranging from 9.58 to 11.43 W/m$^2$ and
14.02 to 15.69 W/m$^2$, and biases values ranging from 0.11 to 0.57 W/m$^2$ and 4.99 to 9.53 W/m$^2$,
respectively. Besides, the Mod2 still exhibited lower accuracy than the other four clear-sky
models, with the highest validated RMSE value of 11.43 W/m$^2$. The performance of Modnew
was the best among the four all-sky models, with the smallest validated RMSE value of 10.76
W/m$^2$ and bias of 3.53 W/m$^2$. Similar to the hourly results under the clear-sky conditions, the
validation results improved considerably if all four all-sky models were trained using the clear-
sky daily samples: their RMSE values and biases decreased to ~8−13 W/m$^2$ and were nearly
zero, respectively, which were even better than the corresponding decreases measured for Mod1
to Mod5. Modnew was the best in comparison to the other three all-sky models, in this case
yielding an RMSE of 8.36 W/m$^2$.
**Table 7**
*Overall Validation Accuracy of the Nine Ocean-surface $R_l$ Models under Clear-sky Conditions at*
*the Daily Scale. The Values in Parentheses for Mod1–Mod8 are the Validation Results Found*
*Using Their Original Coefficients*

| Models | $R^2$ | RMSE(W/m$^2$) | bias(W/m$^2$) |
|---|---|---|---|
| Mod1 | 0.89 (0.90) | 9.66 (11.16) | 0.38 (-2.00) |
| Mod2 | 0.85(0.85) | 11.43 (11.91) | 0.45 (-3.35) |
| Mod3 | 0.90(0.90) | 9.87 (13.57) | 0.11 (9.06) |
| Mod4 | 0.88(0.88) | 10.50 (12.62) | 0.57 (7.16) |
| Mod5 | 0.89 (0.89) | 9.58 (11.92) | 0.39 (6.97) |
| Mod6 | 0.87 (0.88) | 14.32 (18.43) | 9.53 (15.26) |
| Mod7 | 0.87 (0.88) | 14.02 (13.67) | 8.15 (-9.18) |
| Mod8 | 0.80 (0.81) | 15.69 (19.63) | 4.99 (-12.56) |
| Modnew | 0.89 | 10.76 | 3.53 |

In summary, for the ocean-surface $R_l$ estimation under clear-sky conditions, the use of an
all-sky model trained with the clear-sky samples is recommended at both hourly and daily scales.
Modnew performed the best of all nine models when trained with the clear-sky samples, and was
comparable with the other five clear-sky models when trained with the all-sky samples.
Furthermore, our validation results show that the accuracy of Mod2 is not as high as that of other
clear-sky  models that include water vapor variable in terms of RMSE.
4.2.2 All sky
4.2.2.1Hourly scale
Table 8 gives the overall validation results of the all-sky hourly scale ocean-surface $R_l$
from the four models against the independent validation samples with the updated and original
coefficients, respectively.
**Table 8**



*Overall Validation Accuracy of Four Ocean-surface $R_l$ Models under All-sky Conditions at the*
*Hourly Scale. The Values in Parentheses for Mod6–Mod8 are the Validation Results Found*
*Using Their Original Coefficients*

| Models | $R^2$ | RMSE(W/m$^2$) | bias(W/m$^2$) |
|---|---|---|---|
| Mod6 | 0.67 (0.65) | 18.53 (19.84) | 0.05 (3.83) |
| Mod7 | 0.66 (0.64) | 19.06 (26.10) | -0.14 (-10.27) |
| Mod8 | 0.74 (0.51) | 16.91 (37.33) | -0.41 (-28.47) |
| Modnew | 0.76 | 15.95 | -0.04 |

Compared to the results in Table 6, the estimation accuracies under all-sky conditions
shown in Table 8 were generally worse, with lower $R^2$ values (0.66–0.76) and bigger RMSE
values (15.95–19.06 W/m$^2$), which indicates that the uncertainty in the cloud information was the
major reason for the increased uncertainty in the $R_l$ estimation. As in previous results, the three
previous models, Mod6–Mod8, performed much better after recalibration, with decreased RMSE
values up to ~20 W/m$^2$ and their bias values tended to 0; Mod7 still performed the worse.
Modnew performed the best, with an RMSE of 15.95 W/m$^2$ and a bias of -0.04W/m$^2$, followed
by Mod8.

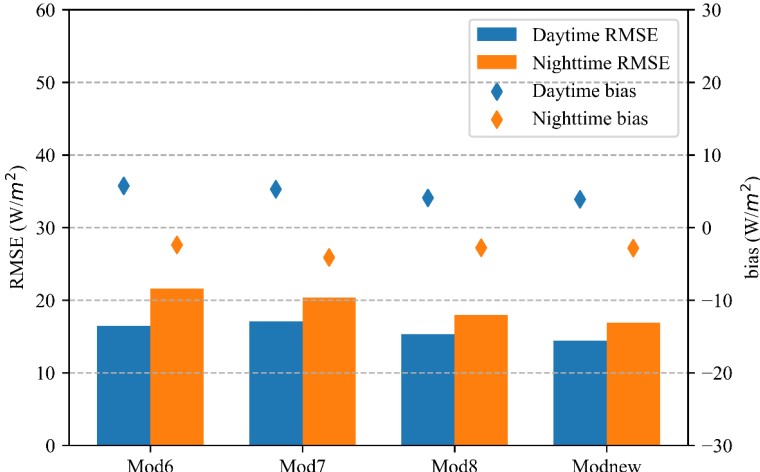

**Figure 6.** Validation accuracy of the estimated Rl under all-sky conditions at the hourly scale for
Mod6-Mod8 and Modnew represented by RMSE (left axis) and bias (right axis).

The hourly results in Table 8 were examined for daytime and nighttime values, as shown
in Figure 6. The results show that the estimation accuracies of the four models were overall
better during the daytime than at nighttime, with smaller RMSE values for the former.
Specifically, during daytime hours, the accuracy of Modnew was similar to that of Mod8, with
RMSEs of 14.43 and 15.33 W/m$^2$, respectively, which were better than those of Mod6 and
Mod7, which yielded RMSEs of 16.46 and 17.09 W/m$^2$, respectively. However, Mod7 performed
a little bit better than Mod6 during the nighttime, although its overall performance was the worst.
It is speculated that the larger uncertainties in the all-sky ocean-surface $R_l$ values at nighttime
can possibly be attributed to the cloud information at nighttime, which was difficult to estimate
accurately compared to the daytime cloud information.





4.2.2.2 Daily scale
Figure 7 shows the overall validation accuracies of the all-sky daily ocean-surface $R_l$
values from the four models. Compared with Mod6–Mod8, Modnew had the best performance,
with an validated RMSE of 10.27 W/m², a bias of 0.10 W/m², and an $R^2$ of 0.88, followed by
Mod8, which yielded an RMSE of 11.96 W/m², a bias of -0.18 W/m², and an $R^2$ of 0.85.
However, Mod8 had a tendency to overestimate low values (<300 W/m²), as did Mod6 and
Mod7.

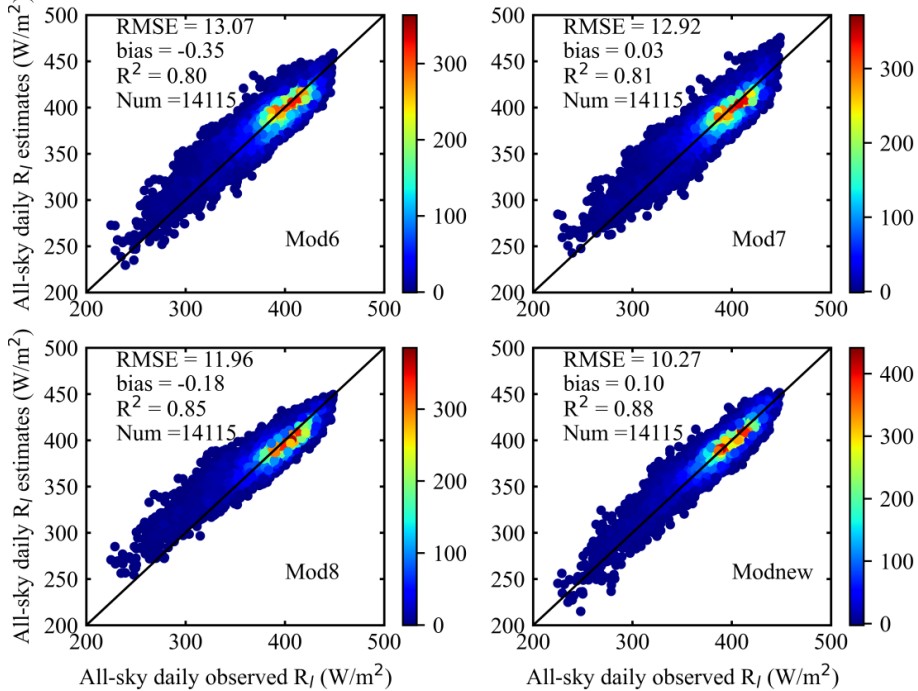


**Figure 7.** Overall validation result of the calculated all-sky daily ocean-surface $R_l$ from the four
models against the independent moored measurements.
Overall, it is speculated that Modnew performed better than Mod6–Mod8 because of the
introduction of two cloud-related parameters (clw and ciw) into the model in addition to the
cloud fraction. In order to demonstrate this speculation better, the relationship between the
estimation errors in the daily all-sky ocean-surface $R_l$ of the four models and clw, which was
used to represent the CBH, was further analyzed. The corresponding mean of the estimation
errors in the daily all-sky ocean-surface $R_l$ and its SEM for each bin of clw in logarithmic format
(in 10% increments) were calculated, as presented in Figure 8.

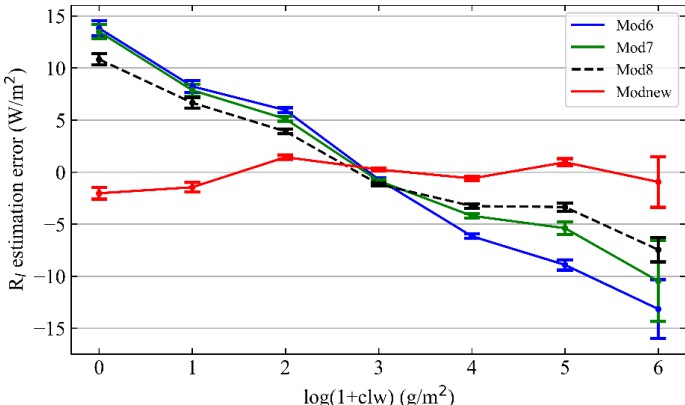


**Figure 8.** The averaged $R_l$ estimation errors and its SEM of Mod6 – Mod8 and Modnew varied
with clw in logarithmic format.

From the results in Figure 8 it can be seen that the $R_l$ estimation errors of Mod6–Mod8
were negative linearly related to increasing log(1+clw); such behavior is not seen for Modnew.
This indicates that the cloud information related to the variations in daily ocean-surface $R_l$ are
not fully characterized by only the cloud fraction. Although Mod8 performed better than Mod6
and Mod7 because of the introduction of the dew point depression to compensate for the
difference between the surface temperature and cloud base temperature, the contributions of the
cloud base emission to $R_l$ still cannot be thoroughly expressed over the ocean surface. Hence,
Modnew performed superior to other models because it also takes clw as input. Moreover, ciw
was also introduced in Modnew to ensure its robust performance at high latitudes.

### 4.3 Further analysis on Modnew

Based on the direct validation results described above, Modnew satisfactorily estimated
the ocean-surface $R_l$ under both clear- and all-sky conditions at both hourly and daily scales.
Hence, further analysis of this new model, such as testing its performance robustness and a
sensitivity analysis, was conducted, and the results are given below.

### 4.3.1 Modnew performance analysis

In order to examine the robustness of its performance, the spatial distributions of the
validation accuracies of the all-sky $R_l$ estimates from Modnew at the moored buoy sites are
presented in Figures 9(a–b) for hourly and daily scales, respectively. Note that the moored buoy
data from which the number of provided validation samples were less than 50 were excluded to
provide a more objective comparison.

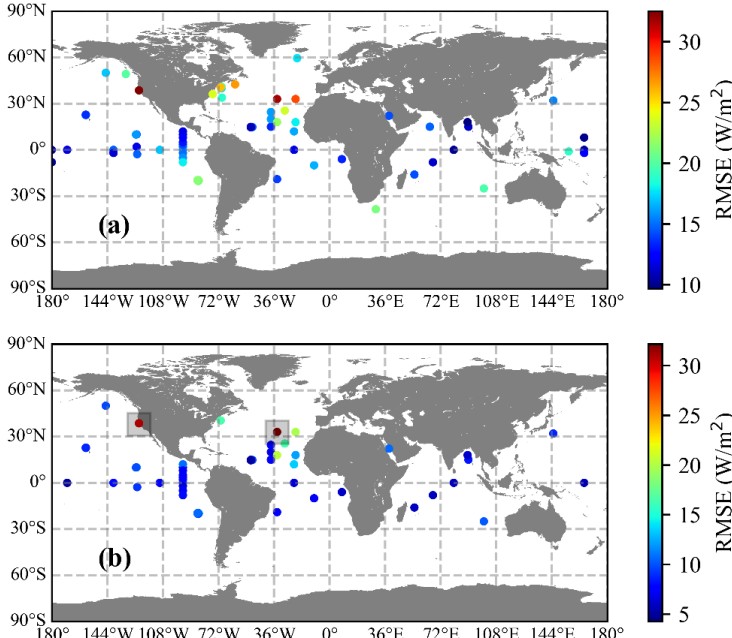

**Figure 9.** Validation accuracies of Modnew on the hourly scale (a) and daily scale (b) at different sites represented by the RMSE values. The two moored buoys in the shaded boxes in (b) are UOP_SMILE88 (38°N, 123.5°W) and UOP_SUB_NW (33°N, 34°W).

The spatial distribution of the validation accuracy (represented by RMSE) of the $R_l$ estimates from Modnew was similar for the hourly and daily data. Their RMSE values got larger from tropical to the high latitude seas, although the daily $R_l$ estimates were generally more accurate than the hourly ones, and the validation accuracy for sites at open seas was more accurate than that within coastal seas. For a better illustration, two time series of the estimated daily ocean-surface $R_l$ from Modnew at two sites were randomly selected and shown in Figure 10, and the one from Mod8 was added for comparison, as well as the corresponding scatter plots. The two buoys, TAO_03 (0°N, 140°W) and OS_PAPA (50°N, 145°W), are in equatorial and mid-high latitude seas, respectively. The temporal variations in the all-sky daily $R_l$ estimates from the two models both captured the variations in the moored $R_l$ measurements very well, but the ones from Modnew were closer to the measurements at high values and low values, especially at the OS_PAPA site. The validation accuracy of Modnew was higher than that of Mod8 at both sites, and Modnew performed better for tropical seas, with validated RMSE values of 6.73 and 10.00 W/m$^2$, respectively, which was assumed that more samples used for modeling were collected at tropical seas and this would influence the model performance at mid-high latitude seas.





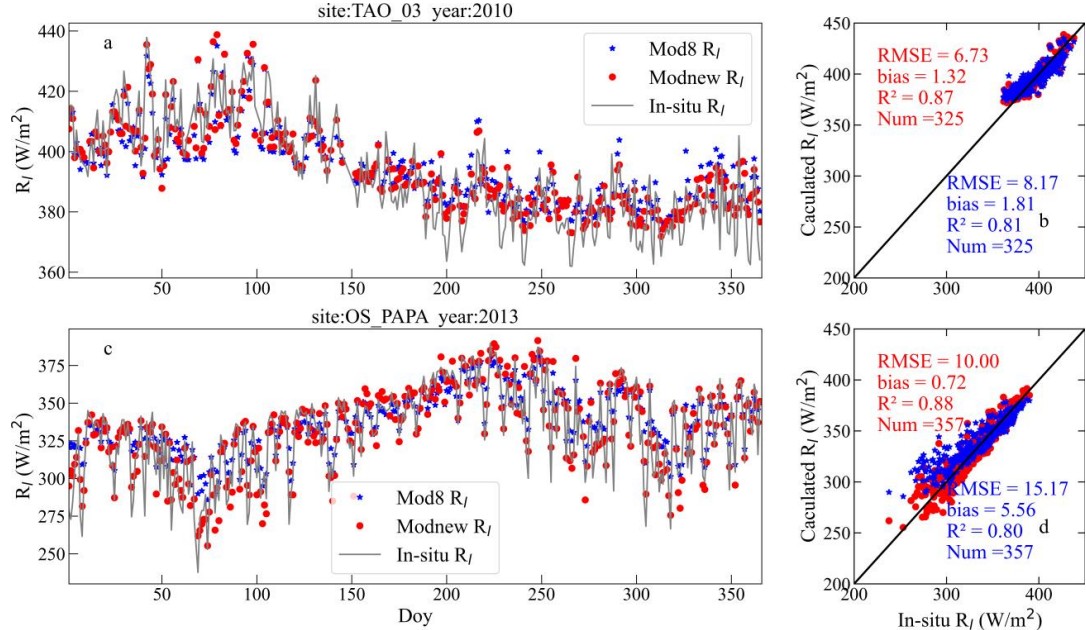

661

**Figure 10.** Time series and scatter plots of the $R_l$ estimates and the moored $R_l$ measurements at
the (a–b) TAO_03 (0°N, 140°W) and (c–d) OS_PAPA (50°N, 145°W) sites. The red points and
blue points represent Modnew and Mod8, respectively.

However, it was noted that Modnew performed poor at some sites, such as
UOP_SMILE88 (38°N, 123.5°W) and UOP_SUB_NW (33°N, 34°W) (see the shaded boxes in
Figure 9). The estimation errors in the daily $R_l$ from Modnew at the two moored buoys were
calculated, as shown in Figure 11, and the ones from the other three all-sky models, Mod6–
Mod8, are shown for comparison. It can be seen that the four evaluated all-sky models all
worked poorly at the two sites, all giving overestimations. A possible explanation may be
attributed to the differences in the characteristics of the atmospheric boundary layer over the two
sites relative to the open sea. Specifically, UOP_SMILE88 is deployed on the northern California
shelf, which is influenced by air temperature inversions (ATIs) (Dorman et al., 1995), and
UOP_SUB_NW is deployed near the eastern flank of the Azores anticyclone system (Moyer &
Weller, 1997). As such, the atmospheric conditions of the two sites are different from those over
the open sea, which would affect the estimation of $R_l$ made with models whose coefficients were
determined by samples collected mostly from sites located in the open sea. Therefore, more
samples should be collected within these seas to help to improve the ocean-surface $R_l$ estimation
accuracy in these areas.





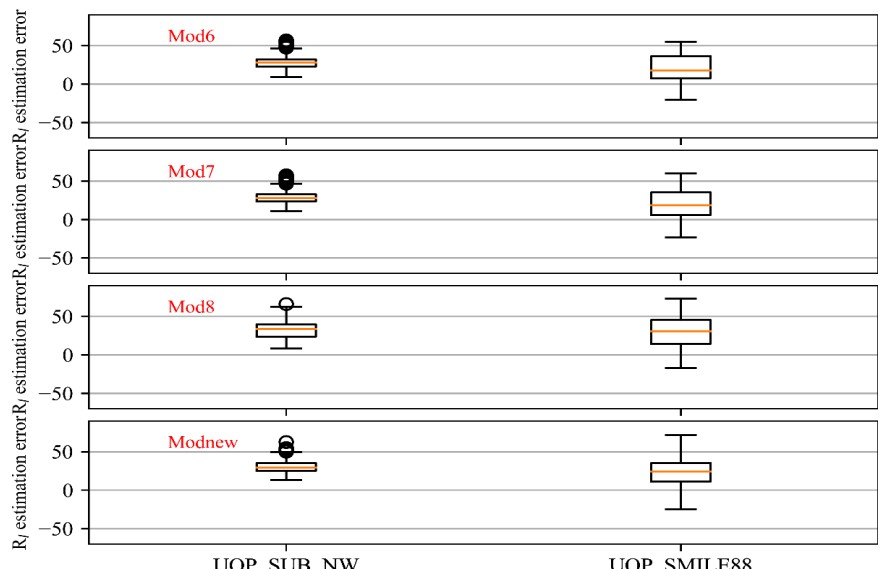

**Figure 11.** Box plots of the $R_l$ estimation errors from models Mod6, Mod7, Mod8, and Modnew at UOP_SMILE88 (38°N, 123.5°W) and UOP_SUB_NW (33°N, 34°W).

### 4.3.2 Sensitivity analysis

In order to quantify the impact of each parameter on the calculated $R_l$ in Modnew, the SimLab software (http://simlab.jrc.ec.europa.eu) was used to conduct a global sensitivity analysis. All inputs in Modnew ($T_a$, RH, C, clw, and ciw) were entered into the software separately, and then 2,000 ocean-surface $R_l$ values were calculated using Modnew by taking 2,000 combinations of these parameters as inputs. Afterwards, the Fourier amplitude sensitivity test (FAST) method (Saltelli et al., 1999) in the SimLab software was employed to conduct a sensitivity analysis based on the inputs, and the corresponding estimated $R_l$ values were used for a sensitivity analysis using the total sensitivity index (TSI). The TSI indicates each parameter's total contribution to the output variance when the interactions of other parameters are also considered, and was used to quantify the sensitivity of each parameter. Table 9 shows the TSI of each parameter in Modnew. Specifically, $T_a$ had the most important effect on $R_l$ with the largest TSI of 41.26%, followed by C (25.6%) and RH (21%). Therefore, the performance of Modnew mainly depended on the accuracy of the $T_a$, C, and RH. The TSI of clw was the fourth highest with 8%, but it is essential to supplement cloud information that cloud cover alone cannot provide, especially for cloud-sky conditions. In terms of ciw, its TSI was just 0.008, which was possibly because only a few samples at high-latitudes were used in this study.

**Table 9**

*FAST Sensitivity Indices of the First Order for Each Input Variable in Modnew*

| $T_a$ | RH | C | Clw | ciw |
|---|---|---|---|---|
| 0.4126 | 0.21 | 0.256 | 0.08 | 0.008 |





## 5 Conclusions

Due to the significance of $R_l$ at the ocean surface, many empirical models have been established for ocean-surface $R_l$ calculation based on observations by relating $R_l$ to some climatic factors, such as $T_a$, RH, and so on. However, most models were developed only for clear days, and for those models that can calculate the all-sky $R_l$, only the cloud cover is taken into account, which is thought to be insufficient for characterizing the influence of clouds on $R_l$, especially for ocean surfaces where cloudy skies are common. Indeed, most previous $R_l$ estimation models were developed only within a specific region based on limited observations, and some for just land surfaces. Consequently, there was a need to perform comprehensive evaluations of these models, including their ability to predict $R_l$ over global seas.

In this study, a new model called Modnew, in which the all-sky ocean-surface $R_l$ is nonlinearly related to $T_a$, RH, C, clw, and ciw, has been successfully developed. This model, as well as eight comparison models, was used to estimate the all-sky ocean-surface $R_l$ at both hourly and daily scales based on comprehensive observations collected from 65 globally distributed moored buoys from 1988 to 2019. In contrast to previous models, Modnew incorporates more cloud-related parameters (i.e., clw and ciw) into the model besides just cloud cover. Modnew and the eight previous $R_l$ models were assessed against the moored values for various cases, including clear- and all-sky conditions at daytime and nighttime and at hourly and daily scales. After careful analysis, several major conclusions could be drawn, as follows:

(1) The eight previous models performed much better after calibration of their coefficients with the global observations for almost all cases, except Mod7 in some situations.

(2) For the clear-sky ocean-surface $R_l$ estimation, the four all-sky models (Mod6–Mod8 and Modnew) could work comparably to or even better than the five clear-sky models (Mod1–Mod5) if their coefficients were calibrated by the clear-sky samples, yielding overall validated RMSE values ranging from 13.40 to 15.40 W/m² at the hourly scale and 8.00–13.00 W/m² at the daily scale. In terms of daytime and nighttime, all five clear-sky models (Mod1–Mod5) performed better at daytime than that at nighttime, and vice versa for the four all-sky models except Mod7. Mod1–Mod5 generally had the tendency to underestimate $R_l$ at nighttime because they do not consider the influence of clouds. Among all models, Modnew was the most robust, yielding RMSE values of 15.03 W/m² and 14.38 W/m² at daytime and nighttime for the hourly scale, respectively.

(3) For the all-sky ocean-surface $R_l$ estimation, the performance of the four evaluated models was generally worse compared to that under clear-sky conditions, which further demonstrated that the uncertainty in the all-sky $R_l$ estimation was highly dependent on accurate cloud information. Specifically, at the hourly scale, the validated RMSE values of the four models ranged from 15.95 to 19.06 W/m², with better performance at daytime. At the daily scale, the RMSE values ranged from 10.27 to 13.07 W/m². Modnew also performed the best in these cases, with an overall validated RMSE of 15.95 and 10.27 W/m² and bias values of -0.04 and 0.10 W/m², respectively. It is worth noting that Modnew performed similarly during both daytime and nighttime at the hourly scale.

In summary, the performance of Modnew was superior to other previous models for ocean-surface $R_l$ estimation for any case, which was mainly because of the introduction of more cloud-related information (clw and ciw). Further analysis of Modnew illustrated the significance of the two parameters as well as cloud cover. However, all results again emphasized that the



accuracy of nearly all the empirical models was highly dependent on the spatial distribution,
quality, and quantity of the samples used for modeling. For instance, Modnew worked better at
open seas in tropical regions where more samples were available compared to other regions.
Therefore, many more samples at different regions, such as in coastal regions and high-latitude
seas, should be collected in the future to improve model performance. Moreover, more accurate
cloud information especially at nighttime is essential to decrease the uncertainty in the estimated
$R_l$ at the ocean surface.
**Competing interests**
The contact author has declared that none of the authors has any competing interests.
**Acknowledgments**
We acknowledge the buoy data sets provided by UOP, GTMBA, and OceanSITES
project. We are grateful to ECMWF for providing the reanalysis data sets. We would also like to
thank the contributions made by the anonymous reviewers and editor that helped improve the
quality of this paper.
**Funding**
This work was supported by the Natural Science Foundation of China (41971291).
**Data availability**
All data sets used in this research, including the moored buoy observations and satellite
and reanalysis data are publicly available. Detailed information on these data sets, including
citations and web links, is presented in Section 3.
**Author contributions.**
PJH and BJ designed and performed the study. All authors contributed to the analysis of
results and final version of the paper.

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
