# Peer review of "Development and Comparison of Empirical Models for All-sky Downward Longwave Radiation Estimation at the Ocean Surface Using Long-term Observations"

_Atmospheric Measurement Techniques, 2024_

## Author Comment (AC1)

This is well-written and a good presentation of a new model for downwelling long wave. As a reader I kept looking for but did not find the motivation for developing the model. I suggest it would improve the paper to explain the motivation.

**Response:** Thank you for your valuable feedback. We agree that clarifying the motivation for developing this model is essential. In short, the main motivation is to address the limitations of existing satellite-based and reanalysis products, which often struggle with accuracy under cloudy-sky conditions and have lower spatial resolution. Our new model is designed to overcome these challenges, providing improved accuracy across both clear- and cloudy-sky conditions, and at finer spatial and temporal resolutions. This makes the model more suitable for localized and high-resolution studies.

Here is the revised version of the first two paragraphs:

[revised manuscript text omitted]

What is the need for a new model? Why not use satellite based long wave radiation at the surface (As Pinker et al describe - this seems to be more accurate than the model you developed.)? Why not use surface radiation from ERA5 reanalysis or from ECMWF model - you import cloud information from ERA5, so why not just take surface radiation from that model?

**Response:** Thank you for this insightful question. The need for a new model arises from several limitations inherent in the existing satellite-based and reanalysis products. While satellite-derived longwave radiation products, such as those described by Pinker et al., generally perform well, they tend to have good accuracy under clear-sky conditions. However, their performance degrades under cloudy-sky conditions, where accurate cloud base temperature and height estimation remains challenging (Zeng et al., 2024). To address this, our model incorporates two cloud parameters—total column cloud liquid water and total column cloud ice water—which help improve the estimation of longwave radiation under cloudy conditions.

Similarly, while products like ERA5 provide surface radiation, they are often generated at coarse spatial resolutions, making them less suitable for local-scale studies or applications requiring finer granularity. Our new model addresses these challenges by providing high accuracy under both clear- and cloudy-sky conditions and across different temporal scales (hourly and daily). The model offers a consistent and computationally efficient method to estimate downward longwave radiation at finer resolutions.

Moreover, we conducted comparisons with ERA5 and CERES longwave radiation products, and our model consistently outperformed these products, including in land regions, where it also surpassed GLASS-AVHRR, ERA5, and CERES_SYN1deg_Ed4A products (Chen et al., 2024).

Years ago, absent satellite or model-based surface radiation, folks needed a model such as your to estimate surface long wave to force, for example, an ocean model. But now people use model or satellite fields. So will anybody utilize your new model? What was the purpose in developing it?

**Response:** Thank you for this comment. While satellite and model-based surface radiation products are widely used today, our model addresses a gap in applications that require higher spatial resolution and greater accuracy for estimating longwave radiation. The proposed model is designed to meet the needs of the atmospheric science and remote sensing communities (Yang et al., 2023; Jiao & Mu, 2022; Zeng et al., 2020; Chen et al., 2024; Zapadka et al., 2020), where parameterization techniques are commonly employed for generating longwave radiation products.

The computational efficiency and high accuracy of our model make it especially well-suited for operational ocean-atmosphere heat budget studies, regional climate modeling, and real-time applications, where existing longwave radiation products are subject to delays. By addressing the limitations of satellite and reanalysis products—such as coarse spatial resolution or reduced accuracy under cloudy conditions—our model provides a valuable and practical tool for researchers and practitioners who need improved, more timely estimates of downwelling longwave radiation.

---

## Author Comment (AC2)

**Response to Reviewer #2's comments:**

**General**: *This study uses data from 65 moored buoys, supplemented with satellite and reanalysis data to evaluate eight existing models for downward longwave radiation Rl, as well as a new model developed here called "modnew". The hourly and daily-averaged Rl estimated from modnew model have overall relatively low errors – very good news as these observations are not widely available. I applaud the authors for gathering so many historic LWR buoy measurements! What a job and what a resource.*

*Response*:
Thank you for your positive feedback and encouraging remarks regarding our study. We are delighted that you recognize the value of our effort in compiling and analyzing this extensive dataset of historic longwave radiation (LWR) buoy measurements. Your thoughtful and detailed comments have significantly enhanced the quality of our manuscript. Below, we provide a detailed response to each of your comments.

**Comment #1**:

*My major concern though is with the universal application of the Pascal & Josey 2000 (PJ2000) longwave radiation (LWR) correction applied to all 65 buoy timeseries.*

*The PJ2000 LWR Correction = (a + lambda)Rsolar + b(Rsolar)^2 ,*

*with a=0.00434, lambda = 0.011, b= 1.72^10^(-6). This is a large correction if Rsolar ~ 1000 W/m2!*

*The polynomial terms involving a & b (a\*Rsolar + b\*Rsolar^2) are a correction for the differential heating of the dome and casing. The last sentence in PJ2000 is "We suggest that such a correction should be made in future analyses if the component temperatures are not logged in order to improve the accuracy of the measured longwave flux". I have emphasized the "if" part here because most if not all of the OceanSITES (including all from NOAA/PMEL, e.g., GTMBA, KEO, Papa, ARC ) LWR sensors are the 3 output Eppley sensors that measure case and dome temperature and correct for this effect. This correction has been done by the data provider and should not be done by the user. Essentially the authors here have double corrected these data.*

*Response*:
Thank you for raising this critical concern regarding the application of the Pascal & Josey (2000) correction across all buoy datasets. We acknowledge that the correction's applicability depends on sensor-specific configurations. Our responses to the key points are as follows:

1. **Prevalence of High Rsolar Values**
   While Rsolar≈1000 W/m2 can occur, such extreme values are rare in our datasets due to typical oceanic cloud cover. Only 1.1% of hourly Rsolar data exceeded 1000 W/m$^2$, limiting the correction's impact.

2. **Data Quality and Pre-Calibration**
   We used "Highest Quality" data, which undergo pre-deployment calibration in the laboratory, as indicated by the quality code definitions (detailed below). While case and dome temperature corrections were not directly available to us, applying a global, consistent correction such as PJ2000 was considered a practical alternative to ensure uniform treatment across all datasets.

```
0 = Datum Missing.

1 = Highest Quality. Pre/post-deployment calibrations agree to within
sensor specifications. In most cases, only pre-deployment calibrations
have been applied.

2 = Default Quality. Default value for sensors presently deployed and
for sensors which were either not recovered, not calibratable when
recovered, or for which pre-deployment calibrations have been determined
to be invalid. In most cases, only pre-deployment calibrations have been
applied.

3 = Adjusted Data. Pre/post calibrations differ, or original data do
not agree with other data sources (e.g., other in situ data or
climatology), or original data are noisy. Data have been adjusted in
an attempt to reduce the error.

4 = Lower Quality. Pre/post calibrations differ, or data do not agree
with other data sources (e.g., other in situ data or climatology), or
data are noisy. Data could not be confidently adjusted to correct
for error.

5 = Sensor or Tube Failed.
```

3. **Magnitude of Adjustment**
   Our analysis revealed that the correction resulted in differences of less than 3 W/m² for 89% of hourly-scale LWR data, within the observational uncertainty of 10 W/m². This suggests minimal over-correction impact.

4. **Purpose of the Data**
   Importantly, the buoy data were not used to validate the LWR products but rather to develop the estimation model. Systematic differences, including potential over-corrections, are accounted for as part of the model's offsets, preserving the integrity of our conclusions.

We appreciate the opportunity to clarify this point and hope this addresses your concern.

**Comment #2**:

*The lambda term, on the otherhand, is intended to correct for solar radiation leakage caused by pinholes or degradation of the dielectric coating on the sensor dome. It is not a universal problem and PJ2000 found that the lambda ranged from 0.007 to 0.024. The 0.011 lambda value is thus a middle of the road case. In some cases, the authors will be adding an error, while in other cases, they will be partially fixing it. Perhaps if the authors keep this correction, it might be also treated as an uncertainty estimate.*

*Alternatively, its more work, but one indication that solar radiation leakage is an issue is if there is a noontime peak in LWR during clearsky days. Perhaps this correction should be applied only in those cases?*

*Response*:
We appreciate your insights into the variability of the lambda term correction. Here is our detailed response:
- The lambda term addresses solar radiation leakage, and its effect varies among sensors. While most sensors exhibit such leakage, the differences between $\lambda$=0.007, 0.011, and 0.024 resulted in minimal changes to LWR, with 78% and 72% of samples showing deviations less than 1 W/m² and 4 W/m², respectively. These values are below the instruments' uncertainty.
- The potential noontime peak in LWR on clear-sky days was considered, but we find that the calibrated differences during non-peak periods remain within the instruments' uncertainty.

Given these results, we maintain that applying a consistent lambda correction introduces negligible bias and does not alter our conclusions.

**Comment #3**:

*The bottom line is that I think the authors should go back and check which sensors are 3-output LWR and then redo the analysis without the a & b terms in the correction for those sites. The authors may also want to review their use of the lambda term correction.*

*Response*:
Thank you for this valuable suggestion. We confirm that we used "high-quality" datasets consisting of raw data that had not been post-processed by the data providers. For these datasets, the Pascal & Josey (2000) correction was applied uniformly to ensure consistency.
Regarding the lambda term correction, we refer you to our detailed response to Comment #2, where we address its impact in our analysis.

**Comment #4**:

*Most of my other comments are to help clarify text, figures, and tables.*

*With the analysis redone with corrections only applied to the subset of observations that do not already have the correction for heating & solar radiation leakage, and with the manuscript revised for clarity, I expect this will eventually make for a well-cited paper.*

*Comments to improve clarity:*

*Table 1 Caption: Add statement that Variables are defined in Table 2.*

*Response*:
Thank you for your suggestion. We have added the requested clarification to the caption. The revised caption now reads:
"Eight Existing Models for Ocean-Surface $R_l$ Estimation, with Variables Defined in Table 2."

**Comment #5**:

*Table 1. Consider expanding this to also include Modnew model. I think this would help the reader find the equation and see its structure in relation to the other models.*

*Response:*
Thank you for the suggestion. Table 1 is intended to present only the existing estimation models for comparison purposes. Since Modnew is a newly developed model introduced in this study, we have chosen to discuss it separately.

**Comment #6**:

*Table 2 show all variables used in this study, including Rl, Rg, DSRtoa, CBH etc.*

*Response:*
Thank you for your suggestion. We have updated Table 2 to include all the variables used in this study, such as $R_l$, Rg, and DSRtoa. However, we have excluded CBH as it was not utilized in our analysis.

**Comment #7**:

*Table 3. In the text, it would be nice if you said where or who these 8 OceanSITES stations are. If 4 of these OceanSITES stations are from TAO (this needs to be clarified), then you only need to describe 4 stations, or even fewer groups as some of these stations (e.g. KEO and Papa) are from one group (NOAA*

*Ocean Climate Stations). These smaller groups making these long OceanSITES time series would benefit from being named in this analysis.*

***Response:***
Thank you for this suggestion. We have clarified the origin of the 8 OceanSITES stations in the manuscript by adding the following sentence:
*"Eight sites from OceanSITES were utilized, specifically: OS_PAPA, OS_KAUST, OS_NTAS, OS_KEO, OS_ARC, OS_JKEO, OS_STRATUS, and OS_WHOTS."*

This ensures transparency and acknowledges the specific sources of these long-term time series datasets.

**Comment #8:**

*Figure 2 caption. What is being represented by the color bar in the left column? What are its units? In the right column, what are the error levels in the "box plots"?*

***Response:***
Thank you for pointing this out. We have clarified the caption for Figure 2 as follows:
*"In the left column, the color bar represents points per unit area. In the right column, the dots indicate the mean value of the $\Delta R_l$ (ME), while the vertical lines represent the standard error of the mean (SEM)."*

**Comment #9:**

*Figure 3 caption. What is being represented by the color bar in a and b? What are the units?*

***Response:***
Thank you for your question. The information in the caption of Figure 3 aligns with the clarification provided in our response to Comment #8:
*"In panels a and b, the color bar represents points per unit area."*

**Comment #10:**

*Figure 4 caption. Same as #6 comment. Also, what is the daytime vs. nighttime criteria?*

***Response:***
We have clarified this in the manuscript as follows:

*"The hourly samples used for independent validation were further divided into daytime (Rg > 120 W/m²) and nighttime conditions (Rg ≤ 120 W/m²)"*

**Comment #11:**

*Figure 7 caption. Same as #6 comment.*

***Response:***
For Figure 7, the color bar represents points per unit area.

**Comment #12:**

*Figure 11. Could you make the y-axis labeling more concise so that it is legible? Also please define what the different levels are in the box plots.*

***Response:***

We have fixed the Figure 11. The top edge, center, and bottom edge of the box represent the 75th, 50th (median), and 25th percentiles, respectively. The whiskers indicate the maximum and minimum values within 1.5 times the interquartile range (IQR), and the circles denote outliers.

[Figure]

**Comment #13:**

*Throughout the text, the observations are described as "screen-level". What does this mean? I've never heard of this.*

***Response:***
Thank you for pointing this out. In meteorology, "screen-level temperature" refers to the air temperature measured at a standard height of 2 meters above the ground.

**Good, E. J. (2016). An in situ-based analysis of the relationship between land surface "skin" and screen-level air temperatures.** *Journal of Geophysical Research: Atmospheres*, *121*(15), 8801–8819. https://doi.org/10.1002/2016JD025318

**Comment #14:**

*Line 41, "Although the ocean-surface Rl is routinely measured at most buoy sites…". Unfortunately, this hasn't been true. Of the 55 TAO sites, only 4 have routinely measured longwave radiation. This is being changed in response to the Tropical Pacific Observing System (TPOS) 2020 project (Kessler et al. 2021), but historically and currently, this statement is only true for OceanSITES bulk flux buoy stations.*

***Response:***
Thank you for bringing this to our attention. We have revised the sentence to remove the word "routinely" to ensure accuracy.

**Comment #15:**

*Line 52. "complicacy" is the wrong word I think. Perhaps "complexity" ?*

***Response:***
Thank you for pointing this out. We have replaced "complicacy" with "complexity" to improve accuracy and readability.

**Comment #16:**

*Line 81 "mid-high" --> "mid-to-high" ?*

***Response:***
Thank you for your suggestion. We have updated "mid-high" to "mid-to-high" for improved clarity and correctness.

**Comment #17:**

*Line 312 "At last" --> "In total" ?*

***Response:***
Thank you for your suggestion. We have replaced "At last" with "In total".

**Comment #18:**

*Line 324 "On the contrary" --> "On the otherhand" ? or "In contrast"*

***Response:***
Thank you for your suggestion. We have replaced "On the contrary" with "On the other hand".

**Comment #19:**

*Line 331. For regions where winds are weak, afternoon near surface stratification can cause the skin temperature to be quite a bit warmer than the bulk SST. This is mainly an issue in the tropics but can also matter in the summer elsewhere. See Cronin et al. (2024) or Clayson and Bogdanoff (2013)*

***Response:***
Thank you for your insightful comment. While we acknowledge that near-surface stratification and skin temperature deviations from bulk SST can occur under low-wind conditions, especially in tropical regions and during summer, this specific issue is beyond the scope of our study, which focuses on downward longwave radiation (Rl) estimation.

We emphasize the importance of addressing such stratification effects in regions with consistently low wind speeds, where autonomous ship-of-opportunity radiometer systems are particularly useful. However, in our dataset, 83% of the samples were observed under wind speeds exceeding 4 m/s, meaning the conditions for significant stratification effects were rare. Therefore, the applied correction remains valid for the majority of our dataset.

Furthermore, **only Mod6** incorporates SST as a model parameter, and its influence is moderated by the Stefan-Boltzmann constant ($\sigma=5.67\times10^{-8}$). As a result, any potential deviations between skin and bulk SST have a minimal impact on the model's performance and do not affect the overall conclusions of our study.

We appreciate the suggested references for further exploration and recognize their relevance to studies focused specifically on stratification effects.

**Comment #20:**

*Line 369, this should reference Table 2, not Table 1.*

*Response:*
Thank you for catching this oversight. We have corrected the reference to point to Table 2 instead of Table 1.

**Comment #21:**

*Line 403. What is CBH ? Perhaps this needs to be included in Table 2.*

*Response:*
Thank you for your comment. In meteorology, CBH typically stands for Cloud Base Height, which refers to the height of the base of a cloud layer above the ground. CBH was not utilized in our study. Therefore, we have chosen not to include it in Table 2.

**Comment #22:**

*Section 4.2.1 Clear sky, is very short. Section 4.2 is Model comparison results, but this 4.2.1 has no results/analysis. How do we interpret these results? Or will that be discussed later? Please let the reader know.*

*Response:*
Thank you for your comment. Section 4.2.1 serves as a subtitle and is further divided into two subsections: 4.2.1.1 (clear sky hourly scale) and 4.2.1.2 (clear sky daily scale). The results and analysis for clear sky conditions are presented within these subsections.

**Comment #23:**

*Line 519. "On the contrary" --> "In contrast"*

*Response:*
Thank you for your suggestion. We have replaced "On the contrary" with "In contrast".

**Comment #24:**

*Paragraph 2 of the Conclusion. Could you use some more words to describe the physical dependencies of the model? This paragraph relies too heavily upon variable names which may be unfamiliar to some readers.*

*Response:*
Thank you for this excellent suggestion. We have expanded the paragraph in the conclusion to include more detailed descriptions of the physical dependencies of the model. The revised text is as follows:

*"In this study, the newly developed Modnew model estimates all-sky ocean-surface downward longwave radiation (Rl) by incorporating key atmospheric and cloud parameters: screen-level air temperature (Ta), relative humidity (RH), fractional cloud cover (C), total column cloud liquid water (clw), and total column cloud ice water (ciw). Ta governs the thermal radiation emitted by the atmosphere, as described by the Stefan–Boltzmann law. RH modifies the atmospheric emissivity by representing the water vapor content. C quantifies the cloud's overall presence, while clw and ciw capture the thermal contributions of liquid and ice clouds, respectively, enabling a more accurate characterization of cloud radiative effects."*

**Comment #25:**

*Paragraph 2 of the Conclusion. Please also clarify how satellite data must be used to run the Modnew*

*Response:*
Thank you for this valuable suggestion. We have expanded the conclusion to clarify the role of satellite data in running the Modnew model. The revised text is as follows:

*"The Modnew model relies on specific atmospheric and cloud-related parameters for accurate Rl estimation. While inputs such as Ta and RH are commonly obtained from in situ measurements, critical cloud-related parameters (i.e. clw and ciw) are typically derived from satellite products or reanalysis datasets, such as ERA5. These parameters are essential for capturing the radiative properties of clouds, which in situ measurements alone cannot reliably provide. Therefore, satellite data or reanalysis products are indispensable for supplying these inputs."*

---

## Author Response (AR2)

*Recommendation: Major Revision*

*My original review recommended that the authors go back, check which sensors are 3-output LWR and redo the analysis without the a & b terms in the correction for those sites that used 3-output LWR. It is very disappointing that the authors decided not to do this.*

*With the greater detail about what LWR timeseries were used, it is clear that all of the time series came from either PMEL or WHOI. All PMEL LWR were 3-output LWR and thus the LWR provided to the public has already been corrected for the differential heating of the case and dome at those sites. I contacted Bob Weller, lead scientist of the WHOI UOP group, and he confirmed that all UOP LWR are also 3-output LWR and thus the user SHOULD NOT apply additional corrections for differential heating to any of the longwave radiation data used in this analysis. The author's post-processing with the "a" and "b" terms in equation 9 should not be referred to as "correction". It is an "adjustment" to the data that actually introduced a bias error.*

*The data providers at WHOI and PMEL do not want users to do this adjustment. In order to discourage any readers and future users from doing a similar "bad practice", the authors have two good choices and one not as good choice:*

*Preferable choice: Explain that all LWR were corrected for differential heating by the data provider and thus the only potential correction needed is for any solar radiation leakage caused by pinholes or degradation of the dielectric coating on the PIR dome. Then reprocess the data using only the lambda term in equation 9. Please provide an explanation for the choice of lambda used.*

*Another good choice: Redo the analysis using the raw data downloaded from the UOP, GTMBA and OceanSITES dataservers. Justification for not applying equation 9 – (a) differential heating correction was already performed by data provider, and (b) the spikes in the LWR associated with the degradation of the dielectric coating on the PIR dome are not seen on all deployments. Therefore a universal application of this correction is probably not appropriate.*

*If the authors continue to use their post-processed data with the 3-term equation 9 adjustment, then, the authors should revise the manuscript to:*
*(1) Clearly state that they learned during the review of the manuscript that the data providers had already corrected all longwave radiation data for differential case and dome heating, and thus the adjustment performed by the authors was not a correction, but instead introduced a bias.*
*(2) Quantify this bias and include it in the error analysis. For hourly data, I estimate the clearsky LWR will have a noon-time bias of up to 5 W/m2. Note that this is much larger than the reported*

*bias in the model fit. For this reason, I recommend that the authors…*
*(3) Remove all results associate with analyses of the biased daytime hourly data, e.g., Figure 3b, Fig. 4, Table 6, Figure 5, Figure 6, etc..*

*Obviously, I very much hope that the authors will choose to redo their analysis with either the raw data provided by the data providers or the lightly-post-processed data that includes a median or mean value of lambda if it is applied universally to all PIR, regardless of indication of solar radiation leakage.*

**Response:**

Thank you for your detailed and constructive feedback.

In response, we have removed the application of the "a" and "b" terms in Equation 9 and reprocessed all longwave radiation (LWR) data using the raw data downloaded from the UOP, GTMBA, and OceanSITES data servers. We now apply no correction for differential heating, as this correction has already been performed by the data providers. We also revised the manuscript accordingly to clearly explain and justify this decision.

As noted in the revised manuscript (Lines 303–313):

*"As pointed out by Pascal and Josey (2000), the main errors in measuring $R_l$ are from the shortwave leakage and differential heating of the sensor. These errors ($\Delta R_l$) in $R_l$ observations can be corrected according to Pascal and Josey (2000). However, this correction was not applied in our study, as (a) differential heating corrections had already been performed by the data providers, and (b) the $R_l$ spikes associated with sensor degradation were not present across all deployments, making a universal correction inappropriate. We also compared the results with and without the correction and found that the conclusions remained unchanged."*

In fact, our updated results indicate that model performance slightly improved without the "a" and "b" adjustments, reinforcing your point that these adjustments introduced a bias rather than improving data quality.

We have accordingly revised all affected figures and tables, including removal or update of results based on inappropriately adjusted daytime hourly LWR data (e.g., Figures 3b, 4, 5, 6 and Table 6).

We appreciate your emphasis on avoiding the propagation of poor practices and believe that the current version of the manuscript aligns well with both the data provider recommendations and community standards.

We thank the reviewer again for pointing out this critical issue, which has led to an improvement in both the accuracy and transparency of our analysis.